# Theory and Practice of Using Pulsed Electromagnetic Processing of Metal Melts

**DOI:** 10.3390/ma15031235

**Published:** 2022-02-07

**Authors:** Nataliya Shaburova, Valeriy Krymsky, Ahmad Ostovari Moghaddam

**Affiliations:** Polytechnical Institute, South Ural State University, Lenin Avenue, 76, 454080 Chelyabinsk, Russia; krymskiivv@susu.ru (V.K.); ostovarim@susu.ru (A.O.M.)

**Keywords:** pulsed electric current (ECP), pulsed electromagnetic treatment (EMP), pulsed magnetic field (PMF), metal melts

## Abstract

In industrial practice, various methods of external influences on metal melts are used. For example, vibration processing, exposure to ultrasound, and other physical fields. The main purpose of such influences is purposeful grinding of the metal structure, which contributes to the improvement of mechanical characteristics. The article presents an overview of research on pulse processing of ferrous and non-ferrous melts: processing with pulsed current, electromagnetic pulses and pulsed magnetic fields. The results of the analysis showed that, despite the different methods and devices used for these treatments, their effect on the structure and properties of the cast metal is generally the same. The main effect is observed in the refinement of the macro and microstructure and a simultaneous increase in the strength properties and plasticity. The intensity of the observed effects depends on the characteristics of the equipment used to create the pulses. The main characteristics are: pulse duration, pulse frequency, current amplitude, and power.

## 1. Introduction

The use of external physical influences for the treatment of molten metals in order to improve the structure and properties of cast metal has been used for several decades. Known works on vibration processing, modification, electromagnetic stirring, and processing by physical fields.

The review considers methods of external pulsed action on metal melts: pulsed electric current, electromagnetic pulses and pulsed magnetic fields. The authors of the works presented in this review use different designations for external methods of influencing melts: for pulsed electric current—ECP, alternating electric current pulse (AECP); for electromagnetic pulse processing—nanosecond electromagnetic impulses (NEMI), electromagnetic pulses (EMP); for pulsed magnetic processing—PMF, pulsed magnetic oscillations (PMO), and low voltage pulsed magnetic field (LVPMF). The overview will use the designations ECP, EMP and PMF for these three modes of exposure, respectively. The considered methods involve the use of equipment with different characteristics and various technological schemes (melt processing in a furnace or out-of-furnace treatment, the use of various equipment, material, shape and size of electrodes). For the convenience of the reader and the possibility of comparing the three exposure options, the material of the article is divided into sections: “Equipment and technologies for pulse processing of metal melts”, which will present all the main options for technological schemes for pulse processing of melts and characteristics of equipment used to generate current pulses, electromagnetic impulses and impulse magnetic field; “Results of the practical use of pulsed processing of metal melts”, which provides actual data on the effect of pulsed processing of melts on the macrostructure, microstructure and properties of ferrous and non-ferrous metals; and in the section “Mechanisms of the impact of pulse treatment on metal melts”, hypotheses are given to explain the effects produced by pulse treatment. The section “Conclusions and outlook” provides a comparative analysis of the literature data on the three pulse processing technologies, identifies their similarities and differences, and also provides recommendations on the possible effective use of such methods of exposure.

## 2. Equipment and Technologies for Pulse Processing of Metal Melts

### 2.1. Processing Scheme of the Pulse Electric Current Processing

The works on the processing of melts with a pulsed electric current (ECP) began in the 1970s–1990s of the 20th century. Figure 1a shows the waveform of the electric current pulses used by Xu et al. [1] for processing metal melts. It can be seen that the pulse profile has a sinusoidal shape and is described by the equation
(1)$$I_{\mathrm{ECP}}=I_{p}\sin\left(\frac{2\pi t}{T_{p}}\right)e^{\left(-1003t-0.6\right)}$$

The main parameters of the pulse are the amplitude of the current strength (*I_p_*), the pulse duration (*T_p_*), the frequency of the pulses (*f*). The amplitude of the current in operation was 1200 A, 1400 A, 1600 A, 1800 A, 2000 A, 2200 A; pulse duration 2.4 ms, 3.9 ms, 4.8 ms, 6.0 ms, 10.2 ms, 11.0 ms; pulse frequency 1 Hz, 5 Hz, 10 Hz, 15 Hz, 20 Hz, 25 Hz.

Figure 2 shows a diagram of the installation for pulsed processing of melts, given in [1]. The low-melting (TL = 10.5 °C) eutectic alloy Ga 20 wt.%; In 12 wt.% Sn (GaInSn alloy) in a cylindrical mold made of plexiglass was subjected to treatment. Two parallel stainless-steel electrodes (8 mm in diameter) were immersed in the melt 10 mm at a distance of 38 mm from each other. In order for the electric current to pass into the melt only through the lower surfaces of the electrodes, the side surfaces of the electrodes were covered with an electrically insulating material (boron nitride). A nylon cover was used to secure the electrodes, allowing vertical movement of the electrodes and, among other things, neutralizing the slight vibration of the electrodes caused by the ECP.

A similar scheme, only with water cooling of the lower part of the mold, was used in [2]. Electric current pulses with a rectangular shape (Figure 1b) were generated by a pe86CWD (plating electronic) power supply. Pulse characteristics: current amplitude 480 A (I0), pulse frequency 200 Hz (f), and pulse duration 0.5 ms (tp). A similar scheme was used in many works, for example, in [3].

A fundamentally different scheme for processing the ECP melt is given in [4] (Figure 3). The side wall of the ceramic mold was wrapped with a resistance wire made of Cr20Ni80 alloy. The mold was placed on a water-cooled copper cooler, which was also used as the bottom electrode. The sidewall was continuously heated to 889 K (616 °C) to prevent radial heat transfer in the mold. The molten metal was poured into this mold, and then the upper electrode, a rod 8 mm in diameter, welded to a rectangular copper block, was immersed in the melt to a depth of 1 mm. To prevent crystallization of the melt, the upper electrode was preheated. The water-cooling system of the bottom of the mold provided directional crystallization from bottom to top. The ECP was generated by a controlled transformer and a silicon rectifier. In the experiments, the current amplitude: 0, 300, 400, and 600 A with pulse frequency 50 Hz were used.

The authors of the above articles and those that will be given in Section 3, as a rule, do not indicate the mass of the processed metal. It can only be roughly estimated based on the size of the equipment used. Calculations show that in all cases the mass of the processed melt does not exceed 0.3 kg, and often much less.

### 2.2. Processing Scheme of the Pulse Electro-Magnetic Processing

One of the first authors who proposed to practically use impulse waves in radar was H.F. Harmuth [5]. A rigorous theory of radiation and reception of pulsed fields was given in the works [6,7,8]. The theory is based on Maxwell’s 6 equations and the classical theory of wave radiation. Subsequently, the field of application of electro-magnetic pulse (EMP) was significantly expanded [9,10,11,12,13,14,15,16,17].

An important feature of electromagnetic pulse processing is the use of unipolar current pulses, positive or negative. The shape of a typical pulse is shown in Figure 4. The pulse duration τ is usually set at half its amplitude. The levels 0.1 and 0.9 set the duration of the leading and trailing edges.

The pulse duration, produced by a noticeable effect on the properties of the metal, is 1 ns. The amplitude of the pulses in the experiments was from 2 to 15 kV. Pulse repetition rate from 100 to 1000 Hz. In most of the studies conducted, the characteristics of pulse duration and amplitude were related to the pulse generator only. Depending on the type of load, they can vary significantly.

Usually, the following basic parameters of pulse generators are set: pulse polarity; amplitude; pulse duration; the duration of the leading or trailing edges; pulse repetition rate; impulse current; pulse energy; output power; type and resistance of the load; time of continuous work; efficiency; supply voltage; cooling method; dimensions and weight. An important characteristic is the cost of the generator.

The duration of the leading or trailing edges affects the rate of change of the current, which can be determined by dividing the value of the pulse current Δj by the duration of the rising edge Δt. It is known [7] that the rate of change of the current Δj/Δt determines the magnitude of the electric field strength E. Assuming that the value of E determines the effect of EMP, then the value Δj/Δt can serve as a criterion for comparing generators. This means that it is desirable to use generators with high impulse currents and short rise times. The quantity Δj depends on the pulse amplitude and can be controlled.

The parameter output power can be estimated in two ways. The first option is the power in one impulse Pp. It can be calculated as the product of the amplitude and the value of the impulse current. For an amplitude of 10 kV and a current of 200 A, the Pp value is 2 MW. The second option may be the average power Pav. To do this, Pp must be multiplied by the pulse duration and by the frequency of their repetition. With a duration of 1 ns and a frequency of 1000 Hz, the value of Pav is 2 W.

For comparison, Table 1 shows the characteristics of generators used for pulsed electromagnetic processing. A distinctive feature of all used generators is the low power consumption of 50–100 W.

Diagrams of installations for processing EMP melts are shown in Figure 5. It can be seen that the main elements of the circuit are: an EMP generator, electrodes (emitters) and a furnace (or mold) with a melt. The shape of the electrodes can be different: a rod, a plate. The main requirement for the electrode material is conductivity and the melting temperature is higher than the heating temperature of the metal melt. We used graphite, copper, and stainless steel. Two connection options are used: in the first (Figure 5a [18]) one electrode was immersed in the melt (the option was tested with and without electrode insulation with a quartz tube), and the second was closed on the body of a conductive crucible; according to the second variant, impulse processing can be carried out not in a furnace, but in a casting mold, in this case both electrodes are immersed in the melt (Figure 5, b [10]). Such installations processed metal melts weighing from several hundred grams to several tons.

Research on the processing of EMP of small masses of metal was carried out by a group of scientists led by Ri Hosen [11]. The scheme used for processing small volumes of metal is shown in Figure 6. The fundamental difference between this method of influencing melts from the above-mentioned connection schemes is the use of the Paraboloid-4 device, which makes it possible to comprehensively investigate the physical properties of melts by the method of penetrating γ-radiation [19]. Steel rods, which were immersed in the melt, were used as the emitter. The mass of the processed metal melts was 250–300 g.

The IYaF generator was used for processing (see Table 1). The metal temperature was recorded using a tungsten–rhenium thermocouple brand VR5/VR 20 [11]. Based on the obtained polytherms of the intensity of gamma-penetrating radiation, the critical points of phase and structural transformations, volumetric changes and thermal compression coefficients of the investigated metals and alloys in liquid, liquid–solid and solid states were determined by the temperature stop and inflections. In parallel, a crystallization thermogram was plotted in temperature–time coordinates.

The use of the “Paraboloid” installation allowed the authors to determine the following crystallization parameters: crystallization temperature (t_sd_); duration of crystallization of aluminum (τ_sd_); thermal compression coefficient during cooling in the temperature range 900—t_sd_.

### 2.3. Processing Scheme of the Pulse Magnetic Field Processing

Pulsed magnetic processing was developed in 2008 (Gong et al. [20] developed a pulsed magnetic oscillatory (PMO) method for refining pure Al structures). In the installation (Figure 7), a capacitor bank with a capacity of 200 μF was used, operating at a voltage of up to 8000 V, followed by its discharge into a copper coil. The current in a coil with a discharge of 2500 V is shown in Figure 7b. The pulse duration, according to Figure 7b is 1 ms, the frequency is 7 kHz. The authors also carried out experiments with the process parameters: amplitude 1500 V, frequency 2 Hz. The location of the crucible relative to the copper coil is shown in Figure 7c.

A similar processing scheme was used Song et al. [21] in Figure 8a. The experiment was carried out in the following sequence: first, a stainless steel rod was placed in a corundum crucible with an inner diameter of 3 mm; second, the sample was slowly heated to a temperature of 1550 °C; third, after 5–8 min of homogenization, the samples were pulled downward at a constant speed and a pulsed magnetic field was applied to them synchronously. Quenching was carried out into a liquid Ga-In-Sn alloy, which made it possible to observe the solid/liquid (S/L) interfaces at stationary stages of growth. The frequency of the pulsed magnetic field was about 4–5 Hz, and the typical discharge mode of the pulsed magnetic field source is shown in Figure 8b. In the course of the experiments, the sample was melted by a heating high-frequency (400 kN) spiral. The length of the section with the molten metal was about 15 mm.

Li et al. [22] in experiments on processing the IN718 alloy melt used: a generator of the pulsed magnetic field, a cooling water system, heat insulation refractory and an alumina mold with an inner diameter of 15 mm and height of 80 mm. The processing was carried out within 5 min. Figure 9 shows the impulse current variation for the PMF based on the result of the inductance–capacitance network (RLC) analysis. For impulse action, three periods can be distinguished: the rise period, the fall period and the off period, which lasts 0.2 s. Rice. Figure 9b shows the pulse profile for one period. The current peaks (50 kA) across the entire coil after a 0.0008 s pulse, and then drops to zero after 0.02 s.

A fundamentally different processing scheme with a magnetic pulsed field was proposed by Yin in 2012 [23]. The experimental setup is shown in Figure 10. It consisted of a pulse generator, an oscilloscope for monitoring electrical pulses and an electric resistance furnace. To transfer a pulse to the melt, a flat spiral-type coil (Figure 10a) made of a copper tube (can be cooled by running water) was used.

## 3. Results of the Practical Use of Pulsed Processing of Metal Melts

### 3.1. Results of the Pulse Electric Current Processing of Metal Alloys

The chronology of studies of pulsed current treatment is well noted in [24]. The use of electric current during metal crystallization was first described in 1974 by Vashchenko et al. [25]. They were the first to establish that electric current has a refining effect on the melt and activates the diffusion of carbon in cast iron [25]. The effect of refinement of the grain structure of pure Al and Al-Cu and Al-Zn alloys upon application of an electromagnetic field at the crystallization stage was noted by Nishimura [26]. In 1985, Misra [27] reported that a direct electric potential applied to the solidifying melt of the Pb—15% Sb—7% Sn alloy changes the process of nucleation and growth of equilibrium phases. Unfortunately, Misra did not provide any explanation for the mechanism of microstructure change [27]. In 1990, Nakada reported a systematic experiment on “modification of solidification structures by pulse electric discharging” [28]. In 1994, Bamak published a short communication and attempted to provide more evidence for the mechanisms of grain refinement caused by applying electric current pulses into solidifying melts [29]. In 1996, Vives reported a systematic study on the “effects of forced electromagnetic vibrations during the solidification of Al alloys [30,31]”. The forced vibration is produced by applying simultaneously a sinusoidal electric current (intensity: i ¼ I sinωt, and frequency: N) and a stationary magnetic field, Bo inside alloy melt contained in a parallelepipedic. Which is similar to the experiments of Nishimura in 1975 [26]. In 2007, Liao et al. conducted experiments to study the effect of ECP during directional solidification on the stability of the liquid–solid interface of the Al—4.5% Cu alloy [32]. In 2014, Rabiger et al. in [33], directional crystallization was studied under the action of strong electric currents on another aluminum alloy (Al 7 wt.% Si). Both direct current and electric current pulses were used. In 2015, on an alloy of the same system, Zhang et al. [34] investigated the formation of segregations of silicon phases during directed crystallization of the Al 20.5 wt.% Si alloy under the action of pulsed electric currents. They found that such an effect causes a strong macrosegregation of the primary silicon phases in the region of the onset of crystallization. The calculations carried out by the authors have shown that a high flow rate (forced convection) occurs in the crystallization region, due to the lower electrical conductivity of the primary silicon crystals. The resulting convection causes mass transfer between the crystallization front and the bulk of the melt, which leads to the n segregation of primary silicon at the crystallization front [34].

Let us describe in more detail the main results of some works.

He et al. [3] modeled the twin roll casting process and investigated the effect of ECP on the structure of a silicon steel ingot. The authors indicate the following characteristics of the processing process: an ECP generator with a capacity of 15 Fi·mF, a pulse discharge frequency of 0–10 Hi·Hz, the peak value of ECP was adjusted from 0 to 15 Ki·A (Fi, Ki and Hi are known constants that represent the parameter of the equipment factors). Silicon steel (Fe 2.18; Si 0.25; Al 0.31; Mn 0.0092; C 0.0047N (wt.%)) was smelted in an induction furnace and overheated to 1650 °C (liquidus temperature is 1520 °C). After holding at 1650 °C for 15 min, the molten metal was poured into a copper mold in 8 s. Under the same cooling conditions, three samples were ECP treated and the other was not treated for comparison. The duration of exposure to ECP with parallel electrodes was 3 min 40 s. The authors note that the effect of ECP significantly changes the size ratio of crystallization zones—the zone of central equiaxed grains increased by 67.5% (while it was practically absent in the untreated metal).

Ivanov et al. [35] simulated the passage of a single pulse with a current frequency of 105 Hz through an aluminum melt at a temperature of 700 °C. The authors considered three electrode systems:“Point to point” for melt processing in a vessel made of non-conductive materials. One submersible electrode, the second point electrode in the bottom of the vessel;“Point to plane” the system is an open lined vessel with non-conductive side walls. The first electrode is submersible, the second electrode is the bottom of the vessel;“Point to walls” consists of an upper immersion electrode and a container with conductive side walls and an insulated bottom.

For the calculation, the capacitive storage voltage was taken U_0_ = 10–50 kV; storage capacitance C = 0.25–5 µF, discharge circuit inductance L = 1–8 µH. The above range is realized experimentally and gives current pulses with Imax from 3.5 to 80 kA and ω ≈ (160–1900) × 10^3^ rad/s. Calculations have shown that the type of electrode system, as well as the strength of the pulsed current, are important technological parameters that determine the nature of the electric vortex flows, the magnetic pressure and the mixing rate of the liquid–metal conductor during the passage of the pulsed current. The most efficient system among those investigated for processing cylindrical conductors is the point-to-point system. Thus, the maximum flow rates (vij) in the central diametrical section and in the skin layer (vΔ) at U_0_ = 30 kV; C = 1 μF; L = 2 μH are: 1.15 × 10^−3^ and 2.4 × 10^−3^ m/s for a submersible non-insulated electrode; 2.14 × 10^−3^ and 2.28 × 10^−3^ m/s for an insulated immersion electrode. The experiments carried out by the authors on the AK7 and AK9 melts (processing parameters: U_0_ = 30 kV; C = 1 μF; L = 2 μH, pulse frequency 2 Hz per minute) provided: a decrease in the average macrograin size from 42 to 36.5 μm, an increase in the ultimate strength from 150 to 190 MPa, relative elongation from 2 to 6% compared to untreated ECP metal. The use of other values of the capacitive storage voltage for processing gave similar, but less significant results.

Zhang et al. [2] investigated the effect of ECP at various stages of crystallization. Alloy Al-7 mas. % Si was treated at various degrees of supercooling relative to the solidus temperature and for different times: 400 sec (continuously during cooling) at 612 °C; 270 sec (before the start of crystallization) at 610.5 °C; 250 sec (before the start of crystallization) at 609 °C; 12.5 sec (during crystallization) at 607.5 °C. Processing parameters: current I_0_ = 480 A, frequency f = 200 Hz and pulse duration t_p_ = 0.5 ms. The greatest effect of grain refinement (from 2.82 mm to 0.84 mm) was obtained with the first treatment option (Figure 11). The authors also found by direct measurements that the degree of supercooling of the melt during crystallization decreases by 1.6 °C.

In [4], an alternating electric current pulse (ECP) was generated by a transformer and a silicon rectifier, which provided a frequency from 2 to 1600 Hz and a current amplitude from 0 to 10,000 A. The experiments were carried out at current amplitudes of 0, 300, 400 and 600 A and a pulse frequency of 50 Hz. The Al-7 wt.% Si was treated. The authors, using five thermocouples, recorded the temperature in different parts of the sample and recorded that during pulse processing, a change in the nature of the temperature change in the upper part of the melt (a decrease in the cooling rate) is observed, while for the melt below the level of 68 mm from the bottom of the crucible, the character of the temperature does not change significantly. Consequently, the authors did not observe explicit mixing of the melt under the action of the Lorentz force. However, significant differences were observed in the nature of crystallization of melts using pulsed treatment, namely, in the upper part of all pulsed ingots, a zone of equiaxed grains of a fine equiaxed zone (FEZ) appeared, while in untreated ingots only columnar crystals were observed. In the microstructure (Figure 12a–d), as the pulse current increases, the transformation of coarse dendritic grains into a fine equiaxed cellular structure is observed.

Zhao et al. [36] are among the few who have devoted their research not only to the impact of the study of the critical structure of the metal after ECP. They determined how the impulse current affects the corrosion resistance of the metal, using the example of aluminum brass HAl52-2 (Cu: 52 wt.%, Al: 2 wt.%, Zn: balance). Processing parameters: frequency of electrical impulses—15 Hz, impulse voltages—500 V and 700 V, respectively, processing time—30 s. After processing the ECP melt, the authors note structure refinement (the average grain size decreased by 50% at 221.3 μm) and a decrease in the amount of γ-phase particles are observed. If at a pulse frequency of 3 Hz, the decrease in the proportion of the γ-phase is insignificant, then at frequencies of 8 and 15 Hz it is 6 and 13%, respectively. Measurements of the microhardness of the ECP-treated and untreated metal also showed differences, with significant hardening observed even at 3 Hz. The microhardness values are 198, 250, 260 and 274 HV, respectively, for 0, 3, 8 and 15 Hz. The potential for free corrosion of samples treated with ECP was higher than that of samples without treatment, and the corrosion current density decreased from 9.51 × 10^−6^ A·cm^−2^ to 1.3892 × 10^−7^ A·cm^−2^ [36]. The thickness of the corrosion layer decreased from 59.2 µm to 50.2 µm [36], while solid and granular corrosion products were evenly distributed over the surface of the samples. In general, the authors note that the corrosion resistance of brass is improved by blocking the diffusion of Zn atoms.

Ma et al. [37] investigated the effect of ECP on the structure and properties of Cu-37.4 wt.% Pb monotectic alloy. Process characteristics was ECP: current peak 800, 1000, 1200, 1400 A, current density 6 × 10^6^ A/m^2^–2 × 10^7^ A/m^2^, voltage 27, 33, 40, 47 V, current frequency—30 Hz, pulse width 20 μs. The authors note the following structural changes: in the initial alloy after crystallization, large, oriented dendrites of the α (Cu) phase and a lead phase, distributed in the interdendrite regions, were found (Figure 13a). As the current peak increased from 800 to 1400 A in the structure, the dendrites of the α (Cu) phase were refined and transformed into flakes; and the predominant orientation of the dendrites disappeared (see Figure 13b–d). The authors also note an increase in the solubility of lead in the α (Cu) phase with an increase in the magnitude of the current peak: 4.98, 5.83, 7.67 and 9.32 wt.%, respectively, for 800, 1000, 1200 and 1400 A. The results of calorimetric analysis (DSC) showed the effect of the current on the melting temperatures of the alloy: for the untreated alloy it was 956.3 °C, for the treated ECP—956.1 °C, 956.3 °C, 955.8 °C and 955.7 °C, respectively, for 800, 1000, 1200 and 1400 A.

Additionally, Ma et al. in [37], an increase in the hardness of the alloy, a decrease in the coefficient of friction and the rate of wear as the current peak increases from 800 to 1400 A (Table 2).

In recent works devoted to ECP, the main focus of the authors is on studying the effect of such an effect on non-metallic inclusions in the melt. Zhang et al. [38] analyzes the refining ability of pulse processing on the example of the industrial magnesium alloy AZ31. The observed changes in the distribution of MgO inclusions were as follows: in the untreated metal, MgO inclusions were distributed almost uniformly along the height of the ingot; in the treated with the electric impulse action, the predominant arrangement of inclusions in the lower and upper parts of the ingot was observed, while the number of inclusions in the center was small. Variation in the pulse frequency showed that the most optimal frequency is 50 Hz. It is at this frequency that the refining effects are most noticeable: due to the pulsed action, the free energy of the system decreases and the MgO inclusions in the magnesium alloy melt are separated and move to the surface, where they are absorbed by the refining agent. In the process of movement caused by the current, an accumulation of inclusions can be observed, due to which their size slightly increases. At the same time, the calculation in MATLab (Version 8.6, MathWorks, Natick, Massachusetts, USA) shows that the tendency towards aggregation between inclusions with a large radius ratio is more obvious. The authors note that the morphology of the inclusions before and after the pulse treatment did not change significantly.

In works [39,40,41,42], it is also shown that ECP can be effectively used for refining melts from non-metallic inclusions with a size of 2–5 μm. The purification of the melt occurs due to the emergence of a force similar to that under the action of which in [4] the movement of crystals of the solid phase occurs.

### 3.2. Results of the Pulse Electro Magnetic Processing

Experiments on pulsed electromagnetic treatment of melts were carried out on pure metals (aluminum, zinc, copper), aluminum and copper alloys, steels, and cast irons. The volume of processed metal varied from 200 g to 2 tons.

For example, in [18], pure granulated aluminum (99.8 wt.% Al, 0.04 wt.% Si, 0.06 wt.% Fe, 0.0015 wt.% Cu, and 0.018 wt.% Zn), zinc (99.98 wt.% Zn; 0.005 wt.% Pb); AlSi7Mg alloy, and AlSi12 alloy were used for experiments. The weight of the experimental melts was: 250g, 300 kg, 3 kg and 120 kg, respectively. Comparison of the structure of aluminum ingots showed that the average area of macrograins along the height of the ingots after pulsed processing changed from 4.8 to 3.4 mm^2^ and 3.3 to 4.4 mm^2^, respectively, for the central and lower parts of the ingots. The density of the metal was 2.682 and 2.695 g/cm^3^, respectively, for the sample without EMP and with treatment. The hardness of the untreated metal was 17 HB, and the hardness of the processed metal was 23 HB. The specific electrical resistance of the metal after exposure to EMP decreases from 0.0277 Ω·m to 0.0273 Ω·m.

Differences in macrostructure are also observed for zinc castings (Figure 14). In the macrostructure of the raw metal near the surface, there is a 4–5 mm section with dispersed columnar crystals (their thickness is 0.1–0.15 mm) and large equiaxed grains with a diameter of 1–1.5 mm prevail in the central part of the casting. In the processed metal, the same crystallization zones are observed, but of another sizes. The surface zone has a width of about 9–11 mm, and the dimensions of equiaxed crystals increase by a factor of 1.5–2.

AlSi7MgFe samples of 3 kg each, which were cast under laboratory conditions, were compared to reveal differences in their microstructure. After EMP treatment, the silicon eutectic phase became more dispersed and acquired the shape of needles with an average length of 25–30 µm; the proportion of this phase in the structure decreased from 30–35 to 10–15%. The silicon particles in the eutectic of the untreated metal had a rounded shape and a size of 5–10 µm. After impulse exposure, the proportion of eutectic along the grain boundaries of the α-phase of aluminum decreased, due to this, the size of these grains increased from 66 to 85 µm and became more equiaxed. The microhardness of the α-solid solution increased by 10–15%, which indicates an increase in the solubility of silicon in it. The increased solubility of elements in the α-phase was also confirmed by the authors of electron microscopy: the maximum concentration of silicon in the α-phase of the parent metal reached 1.36–1.66 wt.%, and the metal treated with EMP—1.48–1.80 wt.% (with an absolute measurement error of 0.04 wt.%). The hardness of the initial specimens of cast specimens of the Al-Si alloy was 51 HB, for those treated with EMP—63 HB. The tensile strength was 170 MPa and 210 MPa, and the elongation was 4.8 and 18.4%, respectively.

The average dendritic cell area of the AlSi12 alloy after pulse treatment is reduced by 1.5–2 times (Figure 15). Changes are observed in the amount of eutectic silicon: in the initial sample it is 7%, in the processed one—9%. Micro X-ray spectral analysis showed that the average solubility of silicon in the α-solid solution of the original sample was 1.45 wt.%, and in the processed sample—1.6 wt.%. Accordingly, the hardness of this phase increased from 83 HV to 97 HV. The ultimate strength and elongation of the original and processed samples after heat treatment of steel are identical and amounted to 292 and 282 MPa, 4% and 2.4%, respectively.

In [43], the results of pulsed treatment of melts of the Al-50Pb, Bi-38Pb, and Bi-18Sn -32Pb systems are presented. The EMP generator had the following characteristics: a pulse amplitude of 10 kV, a leading edge of a pulse of 0.1 ns, a pulse duration of 1 ns, a pulse repetition rate of 1 kHz, and a calculated pulse power of 2 MW. A distinctive feature of the generator was its low power consumption—50 W.

Al-Pb alloys do not form solid solutions, therefore, in the cast ingot, lead (denser) was located in the lower part, under aluminum, which has a lower density. The layers of lead and aluminum in the ingot are structurally inhomogeneous. Globules of the lead phase were present in the aluminum part of the ingot and vice versa. However, the amount and size of these lead globules in the original and processed metal is not the same: 5% and 1–2%, and 10–15 µm and 3–7 µm, respectively. For the inclusion of the aluminum phase in the lead part of the ingot, the same regularities were observed. The width of the interdiffusion zone at the boundary of the two parts of the ingot after pulsed processing decreased from 24.1 to 7.5 μm. In this case, the diffusion layer of lead phases in the aluminum matrix decreased from 21.6 to 6.25 μm, and aluminum in the lead matrix from 2.5 to 1.25 μm.

The progress of more equilibrium crystallization is also evidenced by the results of processing the two-component Bi-38Pb alloy. The structure of the EMP treated and untreated alloys consisted of primary bismuth and eutectic crystals. However, the phase relationship after the impulse processing changed noticeably. Therefore, in the treated alloy, the proportion of primary bismuth crystal was about 10% (when according to the equilibrium state diagram—13%), while in the untreated one—50–55%. The density of the metal treated with EMP increased from 10.42 g/cm^3^ to metal 10.54 g/cm^3^, which also indicates a decrease in the porosity of the metal.

For the Bi-8Sn-32Pb ternary alloy (Figure 16), differences in the number of equilibrium phases were noted: in the structure of the alloy treated with EMP, there were no primary bismuth crystals and the fraction of the double eutectic decreased from 60% to 40% (see Figure 16b). In addition, a decrease in porosity and an increase in density were observed: from 9.78 g/cm^3^ to 9.91 g/cm^3^.

Ignat’ev et al. [44] for processing 300 g of Al-Ti- system alloy (0.86% Ti, 0.032% C, balance Al) used a generator of unipolar electromagnetic pulses with frequency not less than 1000 Hz, with a pulse lasting not more than 10 ^–9^ sec and with power not less than 1 MV. The authors note a decrease in metal porosity after EMP treatment and changes in the amount and location of Al_3_Ti and TiC precipitates. Therefore, the nature of intermetallic Al_3_Ti growth is from acicular with a size of 7–50 μm to polyhedral with a size of 0.5–12 μm.

In [45], an FID generator with the following characteristics was used as a pulse source: the pulse duration was 1 ns, the pulse amplitude was 10 kV, and the pulse repetition rate was 1 kHz. The treatment was carried out before pouring the metal into a ladle with a diameter of 800 mm and a height of 1200 mm. To introduce pulses into the melt, a graphite electrode with a cross section of 15 mm × 15 mm was used. The duration of the impulse action on the liquid alloy was 10 min. The test material was 35L steel, the mass of the processed metal was 1.9 tons.

According to the results of optical microscopy, the original and processed metal had a ferrite–pearlite structure. Moreover, the ferrite phase was arranged in the form of a grid along the boundaries of pearlite grains. The average size of the initial austenite grain in the untreated samples was 700 μm, and in the processed EMP was 450 μm. Despite a significant increase in the total length of the pearlite grain boundary in the irradiated sample, the ferrite phase is also located continuously along all pearlite grain boundaries.

The results of the mechanical tests are shown in Table 3.

Komkov et al. [46] used a GNP-type EMP generator with the following characteristics: the pulse duration was 0.5 ns, the amplitude was higher than 8 kV, and the repetition frequency of the pulses was 1000 Hz. Processing time was 0, 5, 10, 15, and 20 min. As the starting raw material for tin bronze melting in the experiments, the cassiterite concentrate was used (composition of which (wt.%) below: SnO_2—_36%; FeWO_4—_8%; SiO_2_—23% PbS; CuS—12%; Na(Fe, Mg)_4_Al_2_(Si_6_Al_3_B_3_O_27_OH_3_) (schorl)—11%. The effect of pulsed treatment on the properties of the cast Cu+ 6%Sn is shown in Figure 17 and Figure 18.

It follows from the presented data that the thermal conductivity (λ) of bronze irradiated with EMPs in a liquid state for 15 min increases in 2 times, while hardness increases in 1.24 times; the maximum density (8.93 g/cm^3^) is also observed for the 15 min treatment of the melt.

Deev et al. [47] treated Al-Mg-Si alloy. The used generator of electromagnetic waves created pulses with parameters: positive polarity; amplitude up to 15 kV; duration 0.5 ns; pulse repetition rate up to 1 kHz; and the delay of the output pulse relative to the leading edge of the trigger pulse is 120 ns. The following amplitudes were used in the experiments: 0, 5, 10, and 15 kV. According to the structural analysis data, the authors note that pulse treatment affects the crystallization processes, causing a change in the morphology and refinement of the phase components of the alloy, which leads to a denser fine-grained structure. The number of shrinkage micropores and their size are reduced. An increase in the pulse amplitude from 5 to 10 kV has practically no effect on the design parameters.

Ri et al. [48] consider low silicon cast iron containing 3.7% C, 1.0% Si, 0.5% Mn, 0.1% P, and 0.1% S. Hot metal was irradiated at 1350 °C by electromagnetic pulses for 5, 10, 15, and 20 min. Generator of EMP had the next characteristics: positive pulses; pulse amplitude 6000 V with a 50 Ω load; pulse length 0.5 ns at half height; maximum permissible pulse repetition frequency 1 kHz; delay of the output pulse relative to the front of the triggering pulses 120 ns; and maximum current consumed by the generator over the whole supply voltage range no more than 1.7 A at a frequency of 61 kHz. After irradiation, the melt is cooled at 20 °C/min to 500 °C and the intensity I of gamma penetrating radiation is measured (I ≈ 1/d, where d is the density of the melt). Figure 19 shows the solidification parameters of the iron as a function of the irradiation time τirr. With an increase in τirr, the initial temperature of austenite solidification tL increases steadily up to τirr = 20 min (Figure 19a), while the initial ($t_{e}^{in}$) and final ($t_{e}^{f}$) temperatures of eutectic transformation fall. With an increase in treatment time, the initial ($\;t_{A1}^{in}$) and final ($\;t_{A1}^{f}$) temperatures of eutectoid transformation also fall (Figure 19b). The hardness HB of the irradiated iron declines slowly with an increase in τirr to 20 min (Figure 19c). To explain the decrease in hardness with irradiation, the microhardness of the structural components was measured: austenite (pearlite) dendrites at the center and the periphery. An increase in τirr tends to reduce the microhardness of the pearlite relative to that of the unirradiated gray iron by 100 (H50) when τirr = 15 min.

Additionally, authors marked that during irradiation up to 15 min, graphite plates become smaller. This can explain the increased corrosion resistance of cast iron after EMP treatment. Table 4 shows the dependence on the duration of treatment of the mass corrosion coefficient Δm/S and the volume corrosion coefficient K_CO_H_2_.

### 3.3. Results of the Pulse Magnetic Field Processing

The practice of using a pulsed magnetic field to influence a metal melt began with the magneto-oscillation (PMO) method for refine solidification structures of pure Al proposed by Gong in 2008 [20]. The influence of PMO was investigated at various stages of crystallization of the pure Al (>99.7%): (1) continuously during crystallization; (2) at the stage of burying; (3) before cooling to a temperature of 958 K; (4) for 10 s at the stage of active nucleation; (5) during the growth of crystals; (6) at the initial stage of crystal growth; and (7) during the completion of the stage of crystal growth. The experimental setup consisted of a capacitor bank, a trigger device, an electric coil, and a digital oscilloscope for data storage. A capacitor bank with a capacity of 200 μF can provide a voltage of up to 8000 V. An electric current passes through the coil when the capacitor bank is triggered, and is recorded by a digital oscilloscope through an electric current divider.

Jie et al. [49] also carried out work on pure aluminum. Processing was carried out according to a scheme similar to Figure 7. To create the PMF, a coil of 80 loops with a total height of 85 mm was used, connected to a power source. The peak currents used were 300 A, 400 A and 500 A, and the frequencies were 1 Hz, 5 Hz, 10 Hz, 15 Hz and 20 Hz. The exposure duration varied from 0 to 120 s. It is noted that an increase in the frequency of pulses and the magnitude of the current contributes to the grinding of macrograins. The exposure temperatures were 660 °C (metal melting points) and 680 °C. An increase in the duration of exposure at low temperature leads to an increase in the inequigranularity of the structure. This is explained by the fact that at low temperatures, particles of the iron-containing phase are separated along the boundaries of the crystals, which facilitates the process of detachment of crystals from the walls of the mold by moving flows of the melt. Additionally, the authors note the heating of the melt by 13 °C during PMF treatment.

The authors of [21] investigated the effect of pulsed processing by a magnetic field with a frequency of 4-5 Hz and a magnetic induction of 0, 0.5, 0.84, 1.0, and 1.5 T on the structure of stainless steel. The setup scheme used by the authors assumed the melting of a part of the rod (moving at a speed of 12, 24, 64, and 96 μm/s) from stainless steel and the processing of pulsed processing of the molten section of the rod. This processing scheme allowed the authors to study the morphology of the interface between the remelted section of the rod and the initial one. It is shown that at a rod pulling rate of 12 μm/s with an increase in the magnetic induction of the magnetic field, it transforms the growth mode from planar to cellular with an increase in the dendritic front. At a pulling speed of 24 μm/s without a pulsed magnetic field, the growth mode is the growth of a cellular front. When a pulsed magnetic field is applied, a side branch of the cellular crystal begins to appear and gradually develop. At the same time, with an increase in the magnetic intensity, the growth mode gradually passes from the cellular to the growth of the dendritic front. Moreover, the primary spacing gradually decreases with increasing magnetic induction. At a pulling speed of 64 μm/s without a pulsed magnetic field, the growth mode is the growth of a cellular-dendritic front. The application of a pulsed magnetic field of 0.5 T and 0.84 T converts the growth mode into the growth of the dendritic front. A further increase in magnetic induction forces the growth mode to evolve again into the growth of the cell-dendritic (or cell) front. Additionally, the primary spacing gradually decreases with increasing magnetic induction. At a pulling rate of 96 μm/s without a pulsed magnetic field, the growth mode is the growth of the dendritic front, as shown in Figure 20a. When a pulsed magnetic field of 0.50 T and 0.84 T is applied, the growth mode changes from dendritic to growth of the cell front, as shown in Figure 20b,c. With a further increase in the magnetic intensity, the microstructure acquires a complex morphology, as shown in Figure 20d,e. The authors also note that an increase in magnetic induction does not lead to a decrease in the primary gap at a pulling speed of 96 m/s, but, on the contrary, to an increase, as shown in Figure 21b,e.

In [22], the effect of a low-voltage pulsed magnetic field on the crystallization of a nickel-based superalloy (IN718) was studied. The experimental setup for crystallization in a pulsed magnetic field includes: a pulsed magnetic field generator, a water-cooling system, thermal insulation and corundum mold with an inner diameter of 15 mm and a height of 80 mm. The processing was carried out within 5 min. The authors do not indicate the mass of the processed metal. Figure 22 shows photographs of metal microstructures without processing and after pulsed magnetic processing. The authors note significant differences in the structure, the rough dendritic structure of the untreated one changes into relatively small equiaxed grains after the impulse action. The refinement effect of PMF is improved with the decrease of the cooling rate and the melt superheating.

Yin et al. [23] used another PMF deposition technique with an induction coil located above a crucible with a melt. Commercial pure aluminum (99.7%) was processed. The authors point to the following parameters of pulse processing: maximum pulse current (Ip), discharge frequency (fd) and pulse frequency (fp). The discharge frequency (fd) is the number of pulses per unit time, and the pulse oscillation frequency (fp) is the ratio of the rate of change of the pulse during the cycle. These parameters are independent of each other and can be changed independently. Additionally, one cycle of PMF processing can be divided into two stages: 1. discharge (td); and 2. charging (tc). The authors note that the metal melt was processed by PMF with the parameters 75hI·A and 6kI·Hz (here hI and kI, which describe the characteristics of the machine, are the coefficients of the pulse generator and remain constant). The ingot structure without PMF treatment consisted of three distinct zones, and the middle zone with fine equiaxed grains was narrow. Equiaxial grains from the bottom of the ingot (U-shaped equiaxial zone) are formed due to the cooling effect of the graphite mold wall during the casting process. The crystals resulting at the mold wall sank under the action of gravity and natural convection and accumulated at the bottom. After PMO treatment, the central zone transforms from columnar to fine equiaxed grains, while the upper part around the center still consists of acicular columnar grains. The proportion of the area of the equiaxed zone increased from 15% to 69%, and the shrinkage cavity became smaller.

The magnetic field and the generated eddy current had a sufficiently high frequency (103 Hz), so the skin effect was significant—about 8 mm [23]. This is exactly the thickness of the layer with the Joule heating effect.

In 2014, Liang et al. [50] found that during magnetic pulse treatment, the thermal supercooling of the pure Al melt increased, and the temperature gradient from the wall of the mold to the center of the casting decreased due to the convection of the melt caused by the treatment. An increase in the supercooling effect significantly promoted the nucleation of Al crystals, and a decrease in the temperature gradient over the cross section led to uniform crystallization and stabilization of smaller nuclei. All this ultimately led to a refinement of the structure [50].

A detailed study of the impact of PMF on the structure and properties of casting aluminum alloys of the system Al-Si was carried out in work [51]. The experiments were carried out on two-component alloys (6% and 20% wt.% Si) and on AlSi6Cu2 and AlSi9Mg alloys. Parameters of the pulses used: damped sinusoidal signals with a duration (t) of 50–500 μs and a current amplitude (Imax) of 10–100 kA. Such a pulsed-magnetic effect on the melt can be repeated (n) with a time interval between pulses of 0.5–1 s and more and exert a pressure (P) of 1–10 MPa. For processed melts of casting aluminum alloys of the Al-Si system, the skin layer ranged from 1 to 5 mm.

The structure of the Al-6% Si alloy after processing was refined, which led to an improvement in mechanical properties. The greatest effect of hardening was manifested at the discharge energy W = 0.56 kJ and amounted to: by ultimate strength +27%, by relative elongation +136% [51].

Table 5 presents results of influence of PMP on structure and properties of Al-20%Si alloy.

According to the data in the table, with an increase in the number of pulses, not only the size of the Sip crystals decreases, but also their morphology changes from branched morphology to compact, which also contributes to an increase in the mechanical properties of the Al-20% Si alloy.

Noticeable changes in the structure were also observed for the AlSi6Cu2 alloy: the dendrites of the α-phase of aluminum decreased by 23%, the sizes of Si crystals by 10%. At the same time, the number of these phases increased by 1.7 and 1.23 times, respectively.

After PMF, the physical properties of the alloy also changed: the density and electrical conductivity slightly increase from 2.708 to 2.715 g/cm^3^ and from 17.4 to 17.8 MS·m, respectively, for the untreated and processed sample. An increase in the electrical conductivity of the melt also indicates the refinement of its structural components.

## 4. Mechanisms of the Impact of Pulse Treatment on Metal Melts

To explain the effects of external fields on the properties of cast metal, the authors of the works propose various mechanisms and theories, the main ones will be shown in this section.

### 4.1. ECP Influence Mechanisms

Several possible mechanisms have been proposed to understand the causes of grain refining in ECP melt processing. Nakada et al. [28] suggested that, under the action of forced flow in the melt, a force arises that destroys dendrites and leads to grain refinement. Later, Qin et al. [52,53] presented a theoretical model characterizing the process of nucleation under conditions of minimization of the free energy of the system under the action of ECP. Their theoretical analysis was confirmed by experiments. The minimization of free energy during ECP is considered as a driving force for hydrogen removal [54], removal of agminant nanoclusters [55], and improvement of mechanical properties [56]. In addition, Wang et al. [57] was the first to suggest that the enhancement of nucleation during ECP is associated with a decrease in the potential of the outer electric layer of the liquid cluster. Liao et al. [58] showed that the size of the crystals can be drastically reduced by applying ECP at the stage of nucleation, and suggested that grain refinement occurs due to the effect of “crystal rain”. In addition, various mechanisms have been proposed to explain the grain refinement in alloys treated with ECP: a heterogeneous nucleation mechanism, in which the nucleation rate increases due to increased supercooling [59,60], a dendrite fragmentation mechanism, in which the splitting off of dendrite branches is caused by generated pulsed radiation Joule heating [61,62] and due to vibration created by the Lorentz force [28].

He et al. [3] observed significant changes in the size of crystallization zones in the pulsed melt. The authors explain these changes by a phenomenon called “crystalline rain”, when hypothermia occurs at the outer upper surface of the crystallization bath, leading to the formation of a thin solidified crust, on which columnar crystals begin to form and grow deep into the melt. Due to the fact that ECP causes vibrations in the melt, these crystals break off and sinking into the bath of the melt serve as crystallization centers, increasing the proportion of equiaxed grains. Intense cooling prevents these crystals from dissolving in the melt. The authors note that the intensity of the “crystal rain” depends on the frequency of the current pulses.

Since the role of convection in the refinement of the crystal structure of a metal during ECP has been repeatedly shown [63,64,65], the authors of [4] investigated the influence of other factors on the crystallization process. Using low currents that do not cause significant convection, the effect of a change in the free energy of *Ge* was investigated. When an electric current passes through the melt, the free energy of the *Ge* system changes by an amount [32,33,34]:(2)$$G_{e}=-\frac{\mu }{8\pi }\int \frac{j\left(r\right)\cdot j\left(r^{\prime }\right)}{\left|r-r^{\prime }\right|}drdr^{\prime }$$
where *r* and *r*′ are two different positions in space, *j*(*r*) and *j*(*r*′) are the current densities at positions *r* and *r*′, respectively, and *µ* is the magnetic permeability.

In the process of crystallization, nuclei with electrical conductivity different from that of the melt appear on the walls of the mold and on the contact surface of the electrodes and molten metal. The shape of the crystal nuclei affects the distribution of the electric current. Consequently, different free energies of *Ge* correspond to different configurations of crystal nuclei. When the nucleus moves from the central regions of the melt to the side wall of the mold, the chemical free energy and free energy of the interface do not change. The total change in the free energy of the system is equal to ∆*Ge*, i.e., current density varies from *j*_1_(*r*) to *j*_2_(*r*′). The corresponding change in free energy ∆*Ge* can be expressed as [66,67,68]:(3)$$G_{e}=-\frac{\mu }{8\pi }\iint \int \frac{j_{1}\left(r\right)\cdot j_{1}\left(r^{\prime }\right)-j_{2}\left(r\right)\cdot j_{2}\left(r^{\prime }\right)}{\left|r-r^{\prime }\right|}d^{3}rd^{3}r^{\prime }=-\frac{\sigma_{1}-\sigma_{2}}{2\sigma_{1}+\sigma_{2}}kj^{2}V$$
where *σ*_1_ and is the electrical conductivity of melt and crystal nuclei at 889 K (616 °C), respectively, *V* is the volume of crystal nuclei, and *k* is a positive geometric factor. ∆*Ge* will be positive due to the fact that the electrical resistance of the crystals is lower than that of the melt matrix. 

Equation (3) can be used to explain the movement of electrically neutral nuclei in the melt from the center to the lateral surface [69,70]. The electrical conductivity of the melt at 889 K is 4 × 10^4^ S·cm^−1^ [71], and the electrical conductivity of the primary α-Al phase is 10^5^ S·cm^−1^ [72]. The size of the crystalline nuclei of the α-Al phase is 10 times larger than that of the molten alloy. The current density inside the crystal is greater than outside. To minimize the free energy associated with the ECP, the crystals are forced to move from their original location to the center of the ingot by the force F generated by the ECP. The direction of the force is perpendicular to the direction of the electric current (from the center of the mold to the lateral surface) and is axisymmetric for σ_1_ > σ_2_ [69,70]. The direction is reversed when σ_1_ < σ_2_. An approximate expression for the magnitude of the driving force from the electric current to the crystal nuclei is given as [69,70]:(4)$$F=-\frac{\sigma_{1}-\sigma_{2}}{2\sigma_{1}+\sigma_{2}}\frac{d}{f\left(d\right)}\mu j^{2}V=\mu \frac{d}{f\left(d\right)}\frac{\Delta G_{e}}{k}$$
where *d* is the distance from the central axis of the melt along the radius, and *f(d)* increases monotonically, but nonlinearly with increasing *d*. According to Equation (4), *F* is proportional to the change in free energy ∆*G_e_*. This suggests that *F* forces the crystal nuclei to detach from the upper contact surface and side wall into the melt due to ∆*G_e_*.

The authors of [37] explain the effect of ECP on the properties of a monotectic alloy on the basis of the cluster theory. The Cu-37.4 wt.% Pb alloy melt should contain Cu atoms, Pb atoms, Cu-Cu, Pb-Pb and Cu-Pb clusters. In liquid alloys, solvents and solutes are present as positive and negative ions, respectively, due to the difference in electronegativity. In terms of thermodynamic properties, Cu-Pb clusters were more stable than clusters of solutes and solvents [73]. Consequently, when using ECP, solvent ions can form new smaller volumes of clusters with dissolved atoms. As a result of the transformation, the number of Cu-Pb clusters increased, as well as the average force of interaction between the atoms of the solvent and the solute. This led to a decrease in the activity of each component in the metal melt and, therefore, to a decrease in segregation during solidification. Due to the ECP treatment, the number of dissociated individual atoms in the Cu-Pb alloy melt decreased, while most of the atoms were stable in the form of Cu-Pb clusters. ECP has been shown to improve the solid-state solubility of a monotectic Cu-Pb alloy. According to Miedem’s model [73]:(5)$$ln\gamma_{i}=\frac{\alpha_{ij}}{RT}\left[\Delta H_{ij}+\left(1-x_{i}\right)\frac{\partial \Delta H_{ij}}{\partial x_{i}}\right]$$
where *γi* is the activity coefficient; Δ*Hij* is the mixing enthalpy; and *xi* is the molar volume fraction. According to Equation (5), the activity coefficients of Cu and Pb in a metal melt are related to the enthalpy of mixing. Consequently, under the action of ECP, the enthalpy of mixing of the metal melt decreased, and the activity coefficients of Cu and Pb in the metal melt decreased.

### 4.2. EMP Influence Mechanisms

To explain the effect of pulsed electromagnetic treatment on the properties of melts, the authors propose several basic mechanisms.

The authors in [12,13,14,15,16,17,18] consider the following possible factors as the causes of changes in properties: electromagnetic stirring, thermal effect, and mechanical action. Obviously, in one degree or another, each of these factors can affect the structure of the metal by changing the crystallization processes. To determine the contribution of each factor, an appropriate analysis was carried out. Thus, in [74], numerical calculations of the skin-layer depth in a copper conductor were carried out for various pulse durations and shapes, which showed that the skin-layer depth for a 1-ns pulse is several microns. It is clear that such a thin layer cannot cause mixing or heating of the entire mass of metal.

Due to the high power of the pulses used for the EMP—about 1 MW, even in spite of the short duration of the impulse action, one can expect the heating of the melt during the impulse treatment. The calculation of thermal fields, carried out using computer simulation of the process in the MathCad environment [75,76], showed that with a 1 ns pulse with a power of 1 MW over the entire surface of the melt in the crucible, the depth of the heat-affected zone, to which heat propagates when exposed to the pulse, is 0.48 μm. The experiments showed that temperature of the melt during the impulse action increases in 16 °C and decreases almost to the initial value (the increase is 1.42 °C) at the depth of the heat-affected zone. Thus, the influence of the thermal factor during EMP processing can also be neglected.

There are works [77,78,79], in which it is shown that ultrasonic vibrations arise in metal melts when exposed to electromagnetic waves. Mechanical vibrations are excited not only in metal samples [80], but also in liquid melts. In [81], schemes of electrodynamic induction excitation of oscillations in liquid metals are presented and the intensity of ultrasonic oscillations arising in a metal with parameters close to experimental ones is estimated. The non-contact excitation of elastic vibrations in a melt using a constant magnetic field and alternating current is considered in detail. For the case of aluminum melting in an induction furnace in a crucible with a diameter of 300 mm with a constant field of 5 × 10^4^ A·m, it is numerically shown that the vibrational pressure on the melt is 2 atm. This pressure is believed to be sufficient to produce beneficial metallurgical effects. It should be noted that the occurrence of mechanical vibrations in metal samples is possible without the imposition of an external magnetic field [80]. A similar situation is typical for metallic melts [82].

For the theoretical substantiation of the above hypothesis about the mechanism of the influence of EMP on metal melts, let us carry out a comparative calculation of the vibration intensity.

The ultrasonic pressure created in the melt and the displacement of particles in metals for both plane and spherical waves are related by the relation [77]:p = ρcωξ = zωξ(6)
where the product of the density of the metal and the speed of sound ρc = z is the acoustic impedance (resistance); ω—circular frequency (ω = 2πf); ξ—displacement of particles from the equilibrium position.

To determine the vibrational pressure from the impact of EMP, you can use the formula for calculating the wave pressure on the surface:p = E(1 + R)/c(7)
where p is the wave pressure, N/m^2^; E is the power of the incident wave per unit area and unit of time, W/cm^2^; R is the reflectance (R = 0 at full absorption, R = 1 at full reflection); c—wave propagation speed, m/s. The wave propagation speed in molten metals is about 4 × 10^3^ m/s [83].

With pulsed excitation of oscillations, the incident pulse power can be approximately calculated by the formula:(8)$$P=\frac{U^{2}}{r}$$
where r is the characteristic impedance of the cable, equal to 50 Ω; U is the generator voltage equal to 10 kV. Having calculated the value of the incident pulse power by Equation (8) P = 2 × 10^6^ W was obtained. The area of the free surface of the metal in the used crucible with a diameter of 80 mm is 5–10^–3^ m^2^. Consequently, the impulse power per unit area is 4 × 10^8^ W/m^2^.

Substituting the obtained value of the impulse power into Equation (8), Rimp = 1.3 × 10^5^ Pa (or 1.3 atm.) is obtained. This value is similar to those fixed in practice [82] and [84].

In [17], a hypothesis is given about the transformation of electromagnetic pulses into ultrasonic vibrations.

If you place a metal in a constant magnetic field H_0_, then where the current density j differs from zero, the metal is acted upon by the Lorentz force with the density:f_L_ = [j,H_0_]/c(9)

The Lorentz force is the simplest force in nature, leading to the transformation of electromagnetic energy into acoustic energy.

In addition to the Lorentz force, there is, however, a force of a completely different nature. Metal ions and conduction electrons are relatively independent systems. The movement of ions and electrons is different: ions vibrate around fixed equilibrium positions, and electrons move “freely” over distances hundreds (and often hundreds of thousands) times larger than the dimensions of a crystal cell. An electromagnetic wave disturbs the equilibrium between electrons and ions. The energy acquired from the electromagnetic wave is transferred by the electrons to the ions as a result of collisions, but the acquisition of energy and its return are spaced apart in space by an amount of the order of the electron mean free path. The result of this is the emergence of a kind of force dipole: although the force due to the direct action of the electromagnetic wave on the metal at H_0_ = 0 is zero (the metal is neutral), the force density is different from zero. The role of this transformation mechanism is the greater, the larger the “shoulder” of the dipole the mean free path of electrons. The calculations given by the authors show the efficiency of transformation in this case:(10)$$W_{1}\approx \frac{1}{\rho_{M}S}\cdot \frac{\beta }{{1+\beta }^{2}}\approx \frac{s\sigma }{\rho_{M}\omega }.$$

This expression can be compared with the efficiency of electromagnetic acoustic transformations in a metal with a free surface, the expression for which can be written in the form:(11)$$W_{2}\approx \frac{1}{\rho_{M}S}\cdot \frac{1}{{1+\beta }^{2}}\approx \frac{S^{3}\sigma^{2}}{\rho_{M}\omega^{2}}.$$

It can be seen that in the limit β << 1, the transformation efficiency in the case of a fixed surface is β times higher than W for a free surface. For this reason, in conducting liquid metals with an appropriate choice of frequency, a significant increase in the amplitude of elastic longitudinal waves is possible.

The calculation of the efficiency of electromagnetic acoustic conversions in some liquid metals, normalized to the efficiency of electromagnetic acoustic conversions in aluminum at a frequency of *f* = 10 MHz at room temperature, is given in Table 6.(the conversion efficiency in aluminum at room temperature is taken as a unit). An experiment carried out on melts of cesium and mercury confirmed these calculations. The measurements were carried out by the pulse echo method using either two spiral coils for generating and recording sound or using one coil covering a glass crucible with a metal enclosed in it.

The authors note that the method of excitation of longitudinal elastic mechanical waves in liquid metals under the influence of EMP can be useful at high temperatures, when there are problems associated with the creation of acoustic contact and the appearance of oxides on the metal surface, which create significant difficulties.

From Table 6, it can be seen that the effects of changing properties are manifested in all metals. This means that even at low conversion factors (as, for example, for lead and bismuth), the pulse energy is sufficient to obtain new properties of cast metals.

Some hypotheses about the influence of EMP on the properties of melts are given in the works of Ri [11,47]. One of them is based on the cluster model of melts. The cluster theory considers a melt as a combination of two structural components: clusters (microvolumes with an ordered arrangement of particles similar to a crystalline one short-range order structures) and a structureless “disordered” zone separating clusters with a chaotic arrangement of particles, usually more “loose”.

Clusters and a structureless zone are thermodynamically unstable and, as a result of energy fluctuations, they continuously locally transform into each other [85]. The ratio of the volumes occupied by the clusters and the disordered zone is determined by the temperature of the melt and the duration of irradiation of the melt by EMP.

Under the influence of EMP, energy fluctuations can arise in the melt, a change in the structure of short-range order in the arrangement of atoms, a decrease in the size of clusters, a decrease in their life span, and a decrease in the temperature of disorder.

A decrease in the temperature of disorder of the melt causes a change in the degree of compaction and the coefficient of thermal compression of the melt upon cooling, the physical properties of the melt, crystallization parameters, and, ultimately, the physico-mechanical and operational properties of the obtained metal alloys.

A decrease in the proportion of clusters leads to a change in properties: a decrease in viscosity, surface tension, an increase in the solubility and uniformity of the distribution of alloying elements in the liquid phase.

Under the influence of EMP, a decrease in the surface tension (*σ*) at the melt-crystal interface causes a decrease in the values of the critical size of the nucleation centers of crystallization of metals and alloys:(12)$$r_{sd}=\frac{2\sigma MT_{m}}{\rho L\Delta T}$$
where *M* is the molecular weight; *Tm* is the melting point; *ρ*—density; *L* is the latent heat of fusion; and Δ*T* is the degree of supercooling of the crystallizing melt.

EMP treatment of melts changes their energy state, reducing the surface tension at the crystal melt and non-metallic inclusions—crystal interface. All this contributes to the formation of additional crystallization centers and grain refinement.

In the case of metal crystallization on the surface of nonmetallic inclusions (for example, austenite on silicon oxides and other elements), EMP treatment of the melt helps to remove large particles during irradiation of the melt, disperse the remaining “floating” particles, increasing the surface energy. To reduce the surface energy, spontaneous “sticking” of clusters on these inclusions to the critical size of the seed centers is possible. In this case, the crystallographic correspondence of the crystallizing phase with non-metallic inclusions is not required.

Electromagnetic pulses induce currents in the melts. They can be large because of the low electrical resistance of the melt. A moving charge in electric and magnetic fields is acted upon by the Lorentz force:(13)$$\vec{F}=q\cdot \vec{E}+q\cdot \left[\bar{V}\cdot \bar{B}\right]$$
where $\bar{V}$—is the speed of movement of the charge; $\bar{E}$ is the strength of the electric field; *q* is the amount of charge; $\bar{B}$ is the magnetic induction.

The action of the force $\vec{F}$ leads to the fact that the charges are “pressed” against the outer surface. The Lorentz force creates a pressure difference, thereby involving the fluid in motion: from the periphery to the center and vice versa. All this causes an intensification of mass transfer and blocks the growth of dendritic branches. In this case, after some supercooling, bulk crystallization is more likely, which excludes the formation of a zone of columnar crystals and determines a homogeneous fine-grained structure in castings [85].

Irradiation of molten metals with EMP provides the possibility of kinetically easier transfer of atoms from one stable state to another and accelerates the processes of diffusion and dissolution of alloying elements. Due to this, the rate of dissolution of alloying elements increases by more than three times.

A decrease in the crystal lattice parameter and an increase in the amplitude of atomic vibrations from the equilibrium position in copper and aluminum upon irradiation of their liquid phase with EMP for 10–15 min allows us to make the following assumption (Figure 23).

Under the influence of EMP on the liquid phase, the bond energy curve shifts towards a lower value of do. At the same time, apparently, the absolute value of the binding energy Eb decreases. For this reason, EMP irradiation of the melt provides the possibility of energetically and kinetically easier movement of atoms from one stable state to another and accelerates diffusion processes and deactivates clusters.

### 4.3. PMF Influence Mechanisms

In the first work devoted to pulsed magnetic treatment on the melt, Gong et al. [20] first of all note that the effects produced by pulsed magnetic treatment on the melt are similar to the effects produced by a pulsed electric current. This happens, according to the authors, due to the fact that the distribution of the electromagnetic field in the melt is the same as in the case of ECP. The pulsed electric current induced by the PMF has the same width and intensity as the ECP generated by the electrodes, their grinding effect on the melt should be identical. However, unlike ECP, an electrode is not needed for processing. According to the electromagnetic theory, the attenuation of electromagnetic waves with different frequencies in a conductor is different. With regard to the total pulsed magnetic field, the pulse width is much wider than that of the PMF. Thus, the PMF method differs from general pulsed magnetic field treatment by the distribution of the electromagnetic field in the melt, which also results in different forces acting on the melt. The influence of PMF on the crystallization process is manifested as follows. Crystallization nuclei that form at the walls of the mold and the molten metal have a large difference in electrical conductivity, therefore, the density of the induced current through them is different, which leads to a different electromagnetic force acting on them. This is the reason that the crystal falls off the mold wall and moves from the surface to the center of the melt. The electromagnetic force acting on a crystal is determined by the following equation:(14)$$d\vec{f}=V_{c}\vec{B}\left({\vec{I}}_{s}-{\vec{I}}_{l}\right)$$
where $V_{c}$ is the volume of the crystal nucleus, ${\vec{I}}_{s}\;\mathrm{and}\;{\vec{I}}_{l}$ are the density of the induced current through the crystal and melt, respectively. $\vec{B}\;\mathrm{and}\;\vec{I}$ exponentially decrease with distance from the surface deep into the melt (at the depth of the skin layer), and the electromagnetic force acting on the crystal changes in a similar way. At a certain depth, the effect of this force will be negligible, and crystals should accumulate, forming clusters. Further movement of crystals, according to the authors, arises under the action of a sound wave. The main factor determining the number of crystal nuclei in the melt is the shock pressure of the sound wave generated by the vibrations of the melt surface. According to the authors, the pressure on the surface of the melt created by a plane electromagnetic wave is equal to:(15)$$p\approx \frac{1}{4}\mu J_{0}^{2}\left(1-\cos\left(2\omega t-\frac{\pi }{2}\right)\right)$$
where *J*_0_ is the maximum value of the current density on the surface of the melt; *ω* is the frequency of a plane electromagnetic wave. The first term of Equation (15) describes constant pressure on the melt. The second term is the resonant part, which generates sound waves that propagate into the melt. Due to the fact that the size of the crystal nucleus is much smaller than the length of the sound waves, it is difficult for a single nucleus to reflect the sound wave. That is, a single crystal nucleus cannot move inside the melt under the action of sound pressure. However, the formed clusters of embryos of a certain size, comparable to the size of the sound wave, will be able to reflect the sound wave. Under the action of sound pressure, crystal clusters move inside the melt. If the acoustic energy density is very high, then radiation pressure acts on the core of the crystal. This acoustic pressure on the crystal core is twice the average sound energy density when a plane sound wave is fully reflected. The average density of sound energy comes from Equation (15) is [86]:(16)$$\bar{\varepsilon }=\frac{p_{m}^{2}}{2\rho_{0}c_{0}^{2}}=\frac{\mu^{2}J_{0}^{4}}{32\rho_{0}c_{0}^{2}}$$
where *pm* is the peak value of the sound pressure, *ρ*_0_ is the density of the melt and *c*_0_ is the sound velocity in the melt. For the pulse wave, the sound pressure on the crystal nucleus is:(17)$$\Delta p_{s}=2f_{0}\int_{0}^{1/f_{z}}pdt=\frac{2f_{0}\vec{\varepsilon }}{f_{z}}=\frac{\mu^{2}J_{0}^{4}f_{0}}{16f_{z}\rho_{0}c_{0}^{2}}$$
where *f*_0_ is the PMF discharge frequency and *fz* is the PMF oscillation frequency. Since the current oscillation frequency is 7 kHz and the speed of sound in the melt is 1000 km/s, the wavelength of the pulsed sound wave generated by the PMF is tens of centimeters. The wavelength is much larger than the size of the crystal core. Consequently, the sound pressure acting on the core of the single crystal is very small, and the sound pressure in the melt is practically equal to the average density of sound energy, i.e., $\Delta p_{1}=\Delta p_{s}/2$Δ. Thus, clusters of crystal nuclei move inside the melt due to different sound pressures.

To explain the phenomenon produced on the metal structure by a pulsed magnetic effect transmitted by a copper coil, the authors of [23] analyzed the electromotive forces that arise around the coil and their effect on the melt. When an electrical pulse passes through a coil installed above the melt, a pulsed magnetic field (B) is created around the coil and at the same time a pulsed eddy current (J) can be induced in the melt. Since the magnetic field and eddy current generated by the electrical impulse flowing through the coil change at a relatively high frequency (∼10^3^ Hz), the skin effect is significant. According to the calculations carried out by the authors, the depth of the skin layer is 8 mm.

Thus, the pulsed eddy current and the Joule heating effect created by it due to the resistance of the melt are concentrated in the surface layer of the melt with a thickness of 8 mm. Using Joule’s law, equations of free vibrations and the equivalent circuit [87,88], the average power of a single pulse acting on the melt during the unloading stage (td) can be obtained as follows:(18)$$P_{1}=\frac{\lambda R}{t_{d}}{\left(\frac{U_{0}}{\omega L}\right)}^{2}\int_{0}^{t_{d}}\exp\left(-2\delta t\right)\sin^{2}\left(\omega t\right)dt$$
where *P*_1_ is the average power of a single pulse; *δ* = *R*/2*L*, ω =$\sqrt{\left(\frac{1}{LC}\right)-{\left(\frac{R}{2L}\right)}^{2}}$, *U*_0_ discharge voltage, *R*, *L* and *C* are the resistivity, inductance and capacitance of the circuit, respectively, and represents the empirical power transfer coefficient determined by experimental measurement. For pulse parameters 75 kI·A and 6kI·Hz, the average power can reach 10 kW at the stage of discharge. Thus, it is safe to predict that thermal fluctuations will be significant in the surface layer, since the discharge stage is 1% of the entire period.

In addition to thermal vibrations, the authors predict another mechanism of action. When an electric pulse flows through the coil, due to the interaction between the magnetic field B and the eddy current J, the Lorentz force F should appear (Figure 24a,b), which acts on the melt. As is known [89], the direction of the Lorentz force is set by the “left hand” rule. The magnetic field is not perpendicular to the surface of the liquid, but is at an angle to the eddy current, therefore the Lorentz force can be divided into two components: Fx and Fy (Figure 24c). Oscillations of the melt occur due to a change in the direction of the Lorentz force with a change in the electrical impulse. According to Lenz’s law, when the field B decreases, the coil attracts the melt, creating a form of forced convection that is limited within the surface layer due to the shielding effect and the gravity of the melt Fg, as shown in Figure 24e. When the field B increases, the coil repels the melt, forming a different pattern of forced convection (Figure 24e), which can affect the internal fluid due to the transfer of force and the limitation of the form wall, and this forced convection can have a strong effect on the formation of a cast structure.

The appearance of convective flows in the melt during pulsed magnetic treatment of the melt is also noted by the authors of [22]. The results of numerical simulations by the finite element method have shown that under the action of PMF, periodic alternating magnetic force and convective fluxes in the nickel-based alloy are created. Figure 25 shows a graph of the magnetic force in the center of the melt for 20 whole periods and for a single pulse at the initial moment of the pulse appearance and during its decay. It is obvious that the magnetic force has an abrupt change in value during the period of increase and the period of decay of the pulse, which can lead to the occurrence of flows (displacement) of the melt.

Thus, according to the results of numerical simulation by the finite element method, the authors showed the presence in the melt of a periodically variable magnetic force and directional melt flows under the action of PMF. The effect of grinding from a pulsed magnetic field, according to the authors, is mainly explained by the dissociation of nuclei from the wall of the crystallizer by vibration of the melt and subsequent separation of the nuclei by convection of the melt. In addition, forced convection of PMF can lead to fragmentation of dendrites and facilitate the dispersion of the broken dendritic shoulder throughout the melt in the form of new nuclei. Thus, a mass of nuclei is formed in the bulk of the melt. Consequently, the nature of the grinding mechanism in pulsed magnetic processing is to enhance heterogeneous crystal formation.

Pulse processing of stainless steel with a magnetic field with a frequency of 4–5 Hz and a magnetic induction of 0, 0.5, 0.84, 1.0, and 1.5 T [21] reduces the stability of the interface between the solid and liquid phases and promotes planar-cell dendritic-cell transition of the growth regime. That is, the application of a pulsed magnetic field reduces the rate of transition under otherwise identical conditions. According to Trivedi and Kurtz [90], the microstructure of directed crystallization demonstrates a planar-cell–dendritic-cell-planar transition due to competition between different physical processes (the effect of convection is not considered). The diffusion length of the solute is defined as l_*D*_ = *D*/*V*, the thermal diffusion length l_*T*_ = Δ*T*_0_/G and the capillarity length d_0_ = Γ/*T*_0_ (for the alloy), where *D* are the diffusion coefficients of the solute, *V* is the growth rate, and Δ*T*_0_ is the equilibrium freezing interval alloy, *G* is the temperature gradient at the interface Solid/Liquid (S/L) and Γ is the Gibbs–Thomson coefficient. For directional crystallization, the rates of transitions are calculated:For planar-cell transition:
(19)$$V_{c}=\frac{DG}{\Delta T_{0}}=\frac{DGk}{mC_{0}\left(k-1\right)}$$

For cellular-dendritic:


(20)
$$V_{tr}=\frac{DG}{k\Delta T_{0}}=\frac{DG}{mC_{0}\left(k-1\right)}$$


For dendritic-cell junction:


(21)
$$V_{trh}=\frac{D\Delta T_{0}}{a\Gamma }=\frac{DmC_{0}\left(k-1\right)}{ak\Gamma }$$


As Equations (20) and (21) show, a change in the transition rate can be the result of a decrease in the temperature gradient at the S/L interface. Therefore, a decrease in the temperature gradient should lead to an increase in the primary gap at a low growth rate [91,92], which contradicts the experimental results. Thus, as in the case of pulsed magnetic treatment, the experimental results of PMF cannot be explained by changes in the temperature gradient. When a pulsed magnetic field is applied, an induction current arises in the melt. As a result of their interaction in the direction from the edge to the center of the melt, an electromagnetic force arises. The electromagnetic force periodically compresses the melt, which leads to its spreading. With directional crystallization, the melt flow can contribute to the stability of the S/L interface and delay the transition from the planar growth regime to the cellular one [93,94] and from the cellular to the dendritic growth regime [94,95]. Consequently, the facilitation of the passage of the solution by the melt flow also cannot explain the reason for the instability of the interface caused by the application of a pulsed magnetic field. The application of a magnetic field causes a change in the Gibbs free energy of a liquid and a solid. At constant pressure, the change in the Gibbs free energy under the action of a magnetic field can be written as [96]:dG = −S × dT − μ × M × dH(22)
where G is the Gibbs free energy, S is the entropy, T is the temperature, μ is the permeability, M is the magnetization intensity, and H is the magnetic intensity. The temperatures of the beginning and the end of crystallization can change when a magnetic field is applied. The reason for this is the different intensity of magnetization of the melt and crystals. For example, Lee et al. [97,98] found that the temperature of the peritectic phase transformation of the Mn-Bi alloy increases by about 20 °C in a magnetic field of 10 T. Experimental results for stainless steel also suggest that the application of a pulsed magnetic field can increase the liquidus and solidus temperatures [99].

An interesting approach to explaining the mechanism of the effect of pulsed magnetic treatment on metal melts was proposed by the authors of [100]. Using the ANSYS software package (Mechanical APDL and FLUENT), they simulated a pulsed magnetic effect on an Al-5 wt.% Cu alloy melt. The exciting pulse current had an amplitude of 300 A, a pulse duration of 0.002 s, and an interval between pulses of 0.198 s. The authors’ calculations show that the magnetic flux density, eddy current density, Lorentz force and Joule heat fall to zero in the interval between pulses and do not accumulate in each period (Figure 26). Their amplitudes and phases decrease with increasing distance to the side surface of the ingot. The Lorentz force (Figure 26e) during an impulse changes its value from a compression force to a tensile force. Joule heat has two peaks per pulse. A pulsed magnetic field causes forced convection in the melt, and the melt velocity is accompanied by sharp periodic fluctuations. The temperature field tends to be uniform due to the stirring effect of the melt flow. The effect of PMF on fluid flow decreases rapidly after solidification begins, and the final solidification time is almost the same as without PMF.

Simulation of the process of pulsed magnetic processing using COMSOL Multiphysics software was also carried out in [49]. It was shown, firstly, that during the time of a full pulse cycle of 100 ms, about 15 ms, forces act in the melt, and the remaining 85 ms of the alloy is in an unstressed state. Secondly, the forces arising in the melt form four separate areas of liquid circulation, in which the destruction of the crystals formed near the walls occurs, which affects the process of metal crystallization. In addition, the authors carry out calculations of the pressure that occurs in the melt during pulsed processing. It is shown that the value of the emerging pressure in the melt is about 180 Pa.

## 5. Generalization and Discussion

### 5.1. Comparative Analysis of the Technology of Pulsed Processing of Melts

Common to all three methods of pulsed melt processing is that the metal is processed in the state of the melt. The schemes for influencing the melt for pulsed electric current and electromagnetic pulse processing are, in general, the same—mainly a contact scheme is used, when the electrodes from the generator are both immersed in the melt or, one of them is closed on the crucible body. For pulsed magnetic processing, a non-contact circuit is used. As for the characteristics of the applied pulses and equipment, their main parameters are shown in Table 7.

According to the data given in the table, it can be seen that the parameters of pulsed processing by current and pulsed magnetic field are generally very close. These two methods differ only in technological schemes and in the method of introducing pulses into the melt-contact, in the case of ECP, and non-contact, in the case of PMF. According to the electromagnetic theory, the attenuation of an electromagnetic wave with different frequencies in a conductor is different. In terms of the total pulsed magnetic field, the current pulse width is much larger than that of the PMF. Thus, the PMF method differs from conventional pulsed magnetic field processing by the distribution of the electromagnetic field in the melt, which also results in different forces acting on the melt. Electromagnetic processing of pulses differs significantly in duration and amplitude. There is insufficient information in the ECP and PMF documents under review to compare the equipment used to generate the pulses. However, in general, it can be said that similar equipment is used to process a pulsed electric current and a pulsed magnetic field, including a capacitor bank, which ensures the creation of pulses with the required characteristics in the circuit. The generation of current and magnetic field pulses occurs due to the periodic discharge of the capacitors. For pulsed electromagnetic processing, a fundamentally different equipment is used (see Table 7).

It was shown in [9] that the main parameter determining the pulsed field is the value of the current derivative Δj/Δt. It can serve as a criterion for comparing the pulses used. The quantity Δj depends on the pulse amplitude and can be controlled. Table 8 shows the results of calculating the derivative of the current calculated for the three considered methods of processing melts. In almost all the works cited, the value of the current derivative has a value of 10^5^–10^7^ A/s. There is a significant difference when using electromagnetic pulses, for which Δj/Δt = 10^12^ A/s.

### 5.2. Types of Processed Materials and Impact Effects

Despite the wide geography of research on pulsed melt processing, the approaches to the choice of materials are basically identical—basically pure aluminum and alloys based on it are processed, some other low-melting systems and, occasionally, steel. Moreover, as noted above, in the case of ECP and PMF, although the mass of the processed metal is not indicated by the authors, judging by the geometric parameters of crucibles and furnace equipment, it does not exceed 200–300 g. Additionally, only in the case of EMP, the indicated volumes of processed metal vary from 300 g to 1.9 T.

The available sources describe experiments on processing by pulsed fields:ECP: pure aluminum, alloys of the Al-Cu, Al-Zn, Al-Si systems, aluminum bronze HAl52-2, alloys of the Pb-Bi-Sn, Pb-Cu, and Mg-Zn systems.EMP: pure Al and Zn melts, aluminum alloys Al-Si-Mg, Al-Si, Al-Pb, Al-Ti-C, low-melting systems Pb-Bi, Pb-Sn-Bi, tin bronze, 35L steel, and silicon cast iron.PMF: pure Al, alloys of the Al-Si, Al-Si-Cu, Al-Si-Mg systems, stainless steel and IN718 nickel alloy.

The approaches and tasks of analyzing the effects produced by impulse exposure are also different for researchers. It should be noted right away that in most of the works they are limited only to a comparison of the macro and microstructure of the raw and processed metal. The reason for this is the small volume of experimental samples. However, there are works in which both mechanical characteristics and corrosion properties are determined. Table 9 summarizes information on the effect of impulse treatment on the structure and properties of the metal. The magnitude of the manifestation of the observed effects depends on the specific processing conditions, but, in general, the effects produced by these three types of influences are almost identical. Consequently, the lack of manifestation of this or that effect, in each specific case, is a consequence of the fact that experiments on its establishment were simply not carried out.

### 5.3. Mechanisms Explaining Impulse Effects

The similar nature of all three pulsed melt treatments explains the fact that the approaches to explaining the mechanisms of their influence are similar. Table 10 shows a summary of the mechanisms of the impact of pulsed processing on melts. Most of these mechanisms, as shown above, have experimental confirmation of a strictly mathematical description.

Thus, it can be noted that the methods of pulsed melt processing (ECP, EMP, PMF) are identical. Their essence lies in the processing of overheated melt. The only difference is in the equipment used to generate the corresponding pulses.

In this regard, the method of pulsed electromagnetic treatment of melts is more universal, which has shown its effectiveness not only in laboratory conditions, but also in industrial conditions, for example, when processing 1.9 tons of steel. In addition, this method has shown its effectiveness on a wide range of non-ferrous and ferrous metals and alloys.

It should also be noted that since all three methods of pulsed action have already shown their effectiveness on conventional alloys, it makes sense to test their effect on the properties of high-entropy alloys that have become widespread in recent decades. The prospect of such processing is confirmed not only by the positive effects observed on conventional alloys (see Table 10), in particular, by a decrease in the free energy of the system under impulse influences, their effect on the temperatures of the onset of crystallization and a decrease in the enthalpy of mixing [73]. Since many high-entropy alloys are characterized by a rather strong heterogeneity of the structure [101,102], the effect of impulse influences on crystallization processes can also open up new prospects for the manufacture of high-entropy alloys, promoting their homogeneity, reducing the proportion of second phases and precipitates, which, for example, have a negative effect on corrosion properties of high-entropy alloys and resistance to high temperature oxidation, in particular [102,103,104]. Moreover, work in this direction has already begun to appear [105].

## 6. Conclusions

Based on the results of the literature review, the following conclusions can be drawn:Common to all three methods of pulse processing of the melt is that the metal is processed in the state of the melt.Technological processing schemes for three treatment methods are identical. The difference is in the equipment used to generate pulses.Generators for ECP and PMF have similar technical parameters. Generators for EMP are distinguished by a significantly shorter duration of the generated pulses and their large amplitude.To study the effect of three pulse treatments, non-ferrous metal alloys (aluminum, copper) were mainly used. For PMF and EMP, the results of experiments with steels and cast irons are additionally given. A distinctive feature of the EMP method is the possibility of processing large masses of metal (several tons), while experiments on ECP and PMF were carried out on small volumes of metal.Common to all three mechanisms of action are such effects produced on the metal as: changing the crystal structure of the ingot (increasing the proportion of equiaxed grains), grinding macrograins; increase in liquidus temperature; refinement of micrograins; increase in yield strength, increase in elongation; increase in yield strength, increase in elongation; and decrease in electrical resistivity. Such effects as: change in the quantitative ratio of equilibrium phases; increasing macro/microhardness; increased corrosion resistance; increasing the solubility of the second component in the main phase; reducing the number of inclusions and dispersed particles of the second phase are noted for ECP and EMP treatments.The following are noted as the main mechanisms of impulse action on the melt: the emergence of force flows in the melt, which cause fragmentation of the formed dendrites and increase the number of crystallization centers and appearance in the melt under the influence of external action of forces of a mechanical nature (vibrations and ultrasonic vibrations).The authors suggest the efficiency of using pulsed effects on the properties of high-entropy alloys.

## Figures and Tables

**Figure 1 materials-15-01235-f001:**
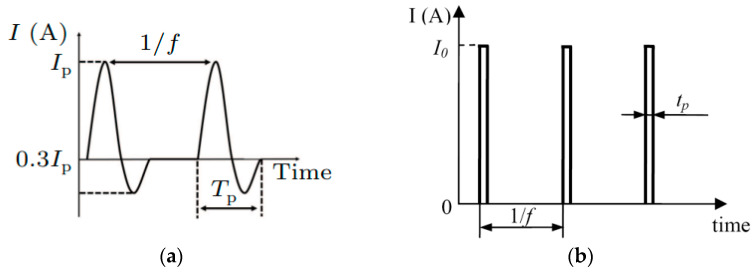
Schematic views of waveform of electric current pulse: (**a**) presented in work [1]; and (**b**) presented in the work [2].

**Figure 2 materials-15-01235-f002:**
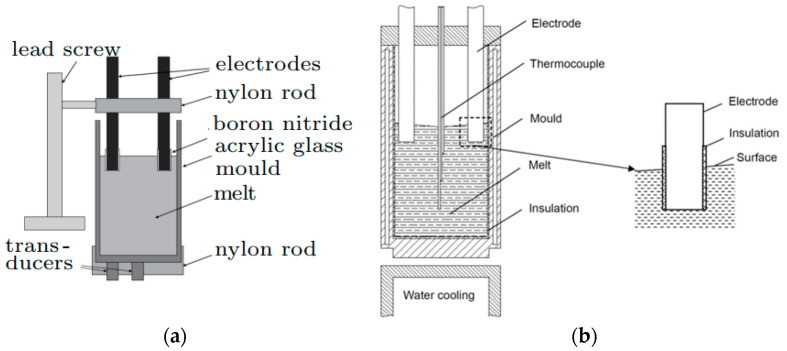
Schematic drawing of the experimental setup: (**a**) [1]; and (**b**) [2].

**Figure 3 materials-15-01235-f003:**
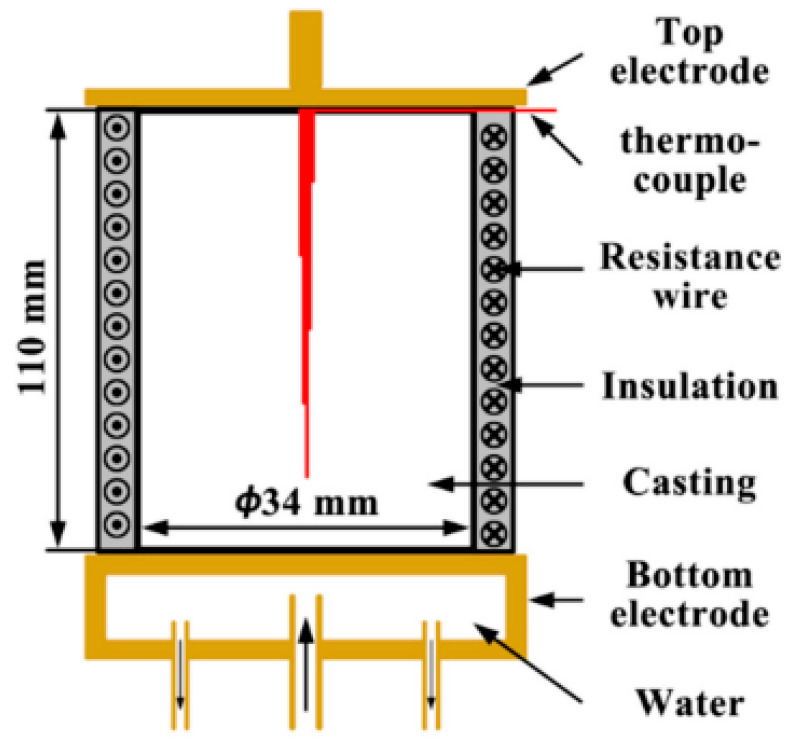
Scheme of the experimental apparatus [4].

**Figure 4 materials-15-01235-f004:**
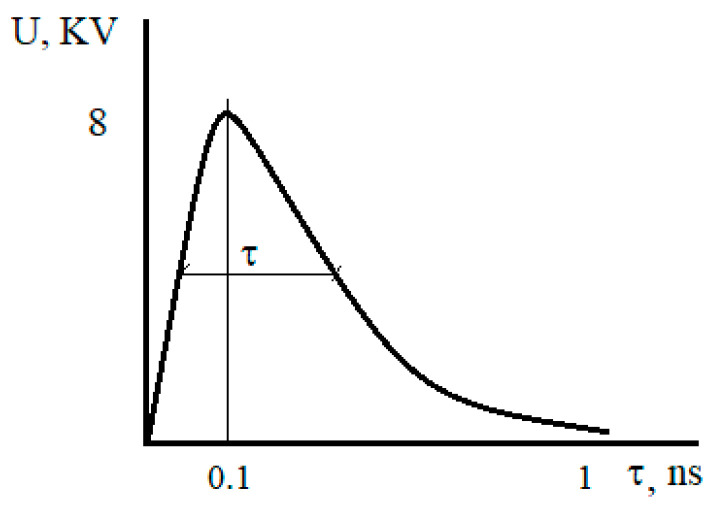
Shape of an electromagnetic pulse [9].

**Figure 5 materials-15-01235-f005:**
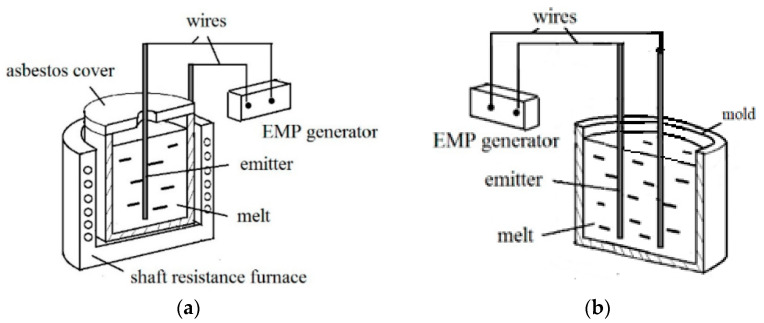
Schemes of processing melts: (**a**) in the furnace [18]; and (**b**) in a mold [10].

**Figure 6 materials-15-01235-f006:**
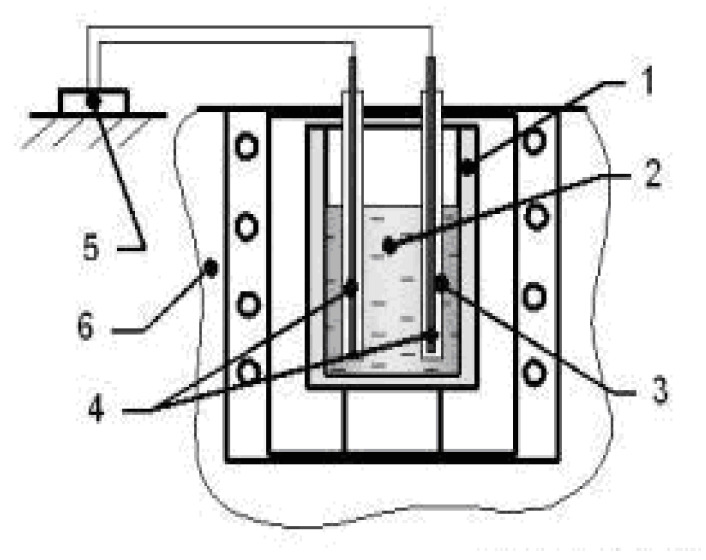
Scheme of treatment of the metal melt with EMP: 1—crucible, 2—melt, 3—alundum tip, 4—electrodes, 5—EMP generator, 6—“Paraboloid-4” installation [11].

**Figure 7 materials-15-01235-f007:**
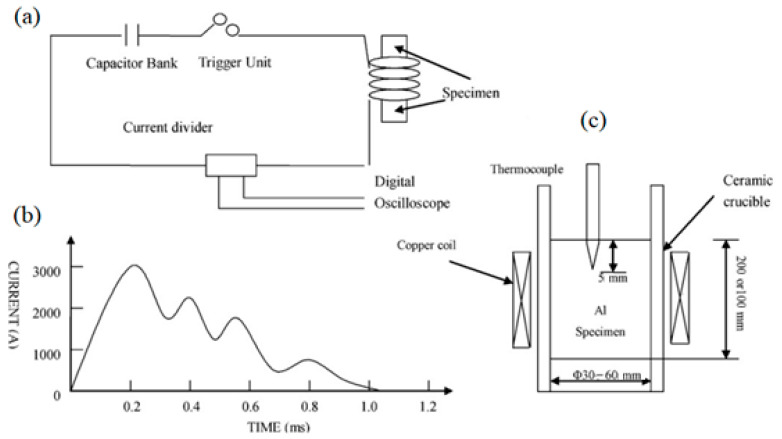
(**a**) Installation schematic diagram. (**b**) The current pulse’s profile (when discharging a 2500 V current from the capacitor bank). (**c**) Scheme of the setup for magnetic field treatment melts [20]. Copyright 2008, Elsevier.

**Figure 8 materials-15-01235-f008:**
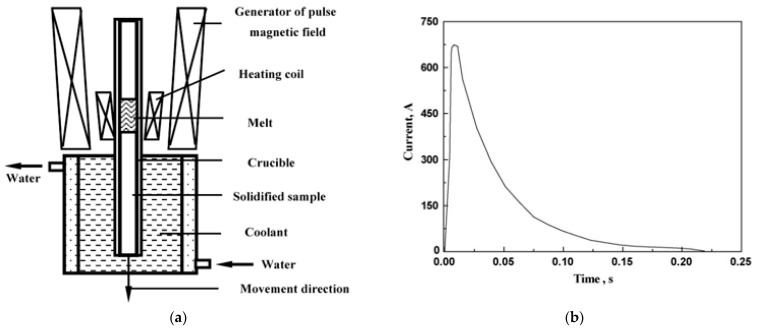
(**a**) Scheme of experimental setup; and (**b**) a discharge course of power supply of pulsed magnetic field [21].

**Figure 9 materials-15-01235-f009:**
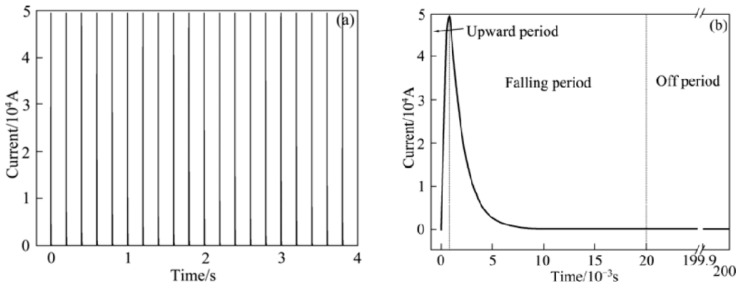
Variation of pulsed electricity in coil: (**a**) 20 whole periods of current; and (**b**) pulse’s profile in one period of pulse [22].

**Figure 10 materials-15-01235-f010:**
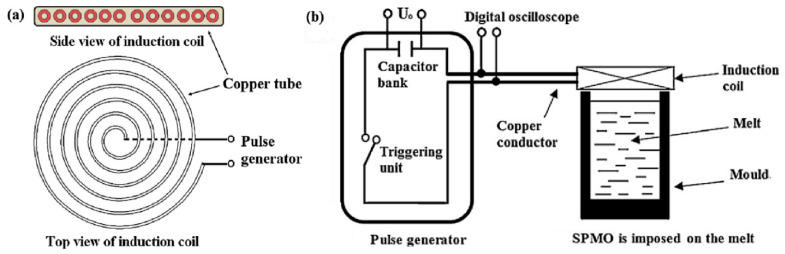
(**a**) Coil appearance, and (**b**) the scheme of the experimental setup [23]. Copyright 2012, Elsevier.

**Figure 11 materials-15-01235-f011:**
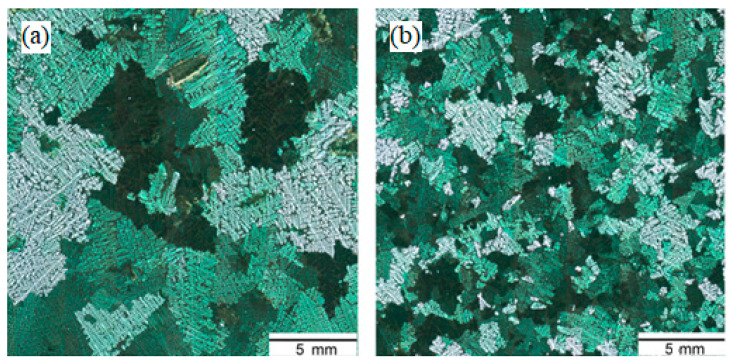
Macrostructure of the test samples: (**a**) without ECP; and (**b**) with ECP [2].

**Figure 12 materials-15-01235-f012:**
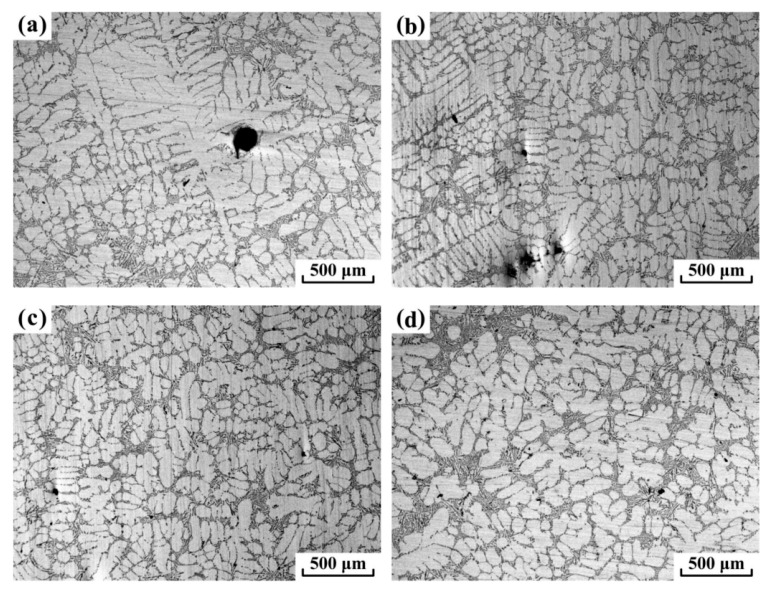
Microstructure of Al-7Si alloy samples: (**a**) without ECP; (**b**) under ECP with 300 A; (**c**) under ECP with 400 A; and (**d**) under ECP with 600 A [4].

**Figure 13 materials-15-01235-f013:**
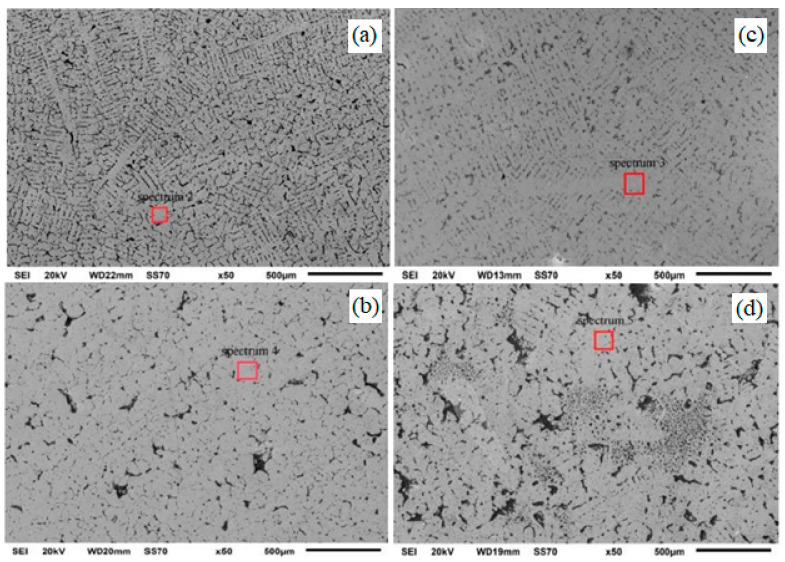
Solidification structure of alloy, treated with different pulse current: (**a**) I = 800 A; (b) I = 1000 A; (**c**) I = 1200 A; and (**d**) I = 1400 A [37].

**Figure 14 materials-15-01235-f014:**
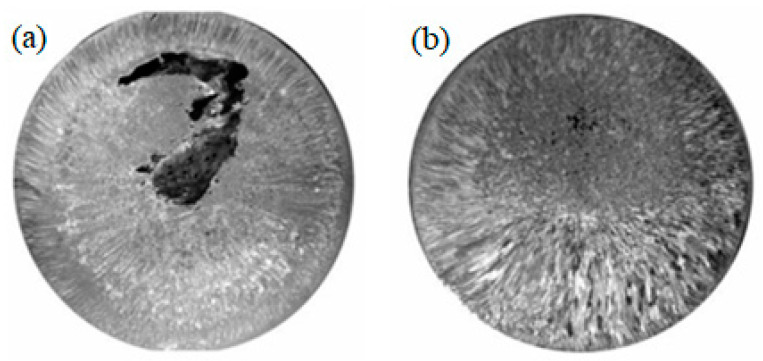
Macrostructure of the templates of pure zinc: (**a**) untreated sample for comparison, and (**b**) EMP processed sample [18].

**Figure 15 materials-15-01235-f015:**
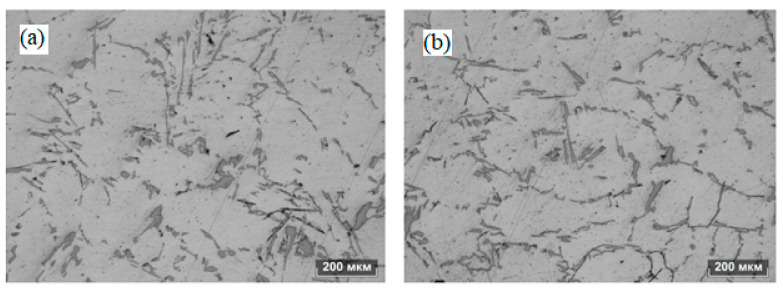
Microstructure of the initial (**a**) and EMP treated (**b**) AlSi12 alloys [18].

**Figure 16 materials-15-01235-f016:**
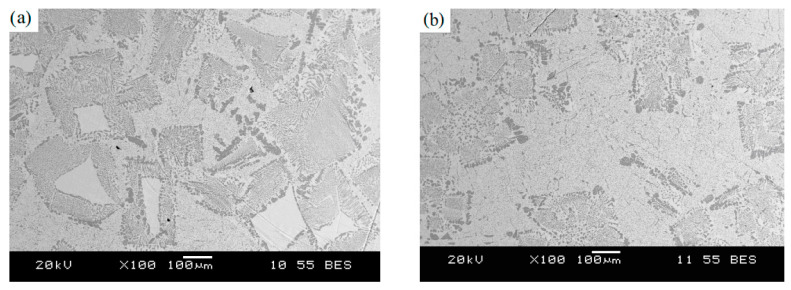
Microstructure of Bi–18Sn–32Pb alloy (**a**) initial and (**b**) with EMP treatment [43].

**Figure 17 materials-15-01235-f017:**
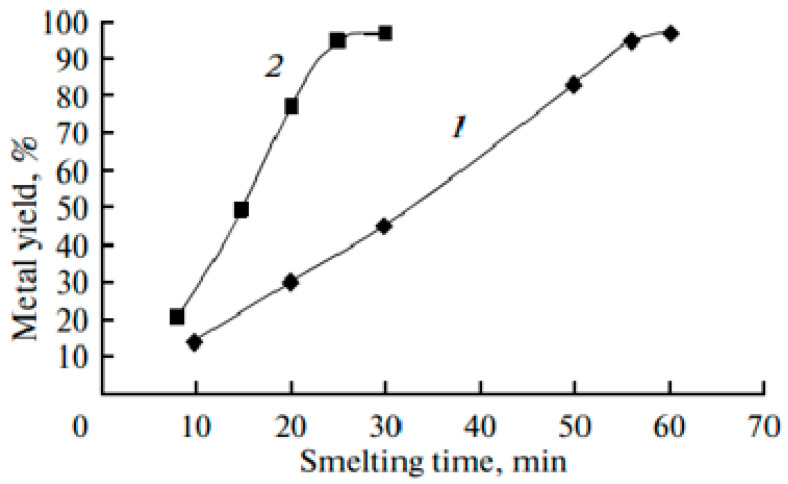
Dependence of the metal yield on the smelting time. (**1**) Without irradiation and (**2**) with irradiation [46].

**Figure 18 materials-15-01235-f018:**
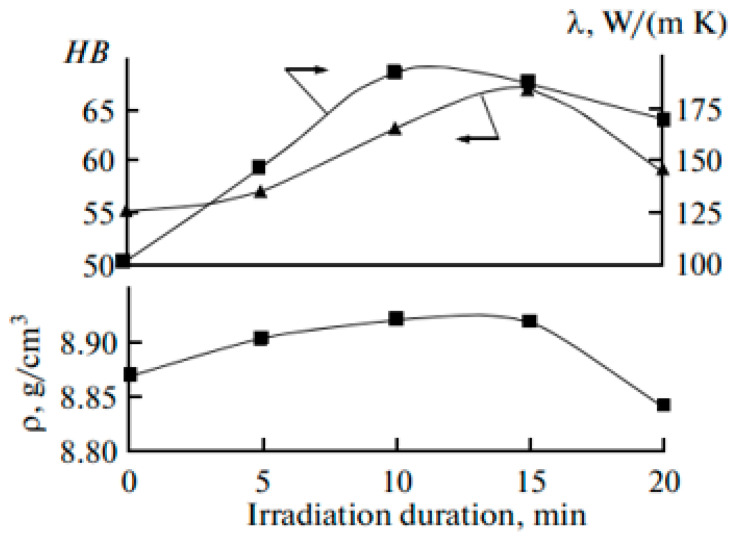
Dependence of the physical–mechanical properties of tin bronze on the EMP irradiation duration of its melt [46].

**Figure 19 materials-15-01235-f019:**
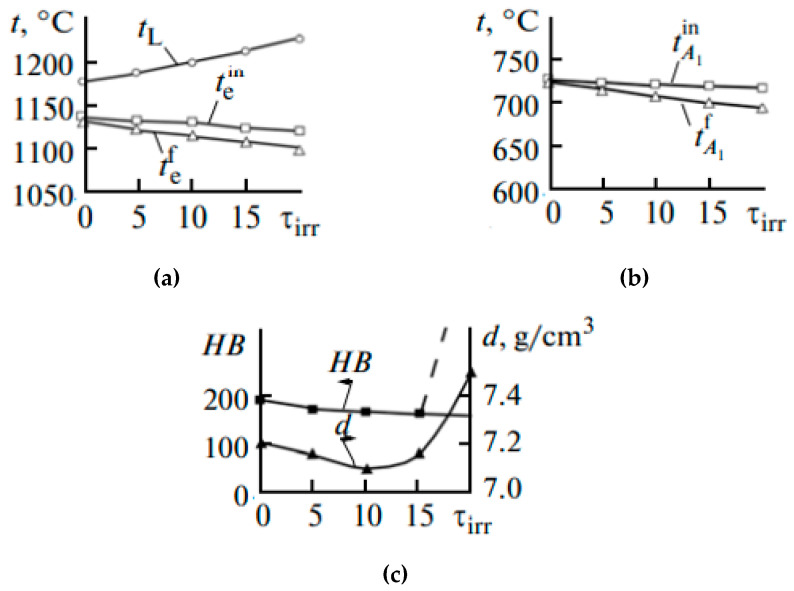
Influence of the irradiation time (τirr, sec) by electromagnetic pulses on the metal properties [48]: (**a**) the temperature of crystallization, the beginning and end of the eutectic transformation; (**b**) the beginning and end of the eutectoid transformation; (**c**) hardness and density.

**Figure 20 materials-15-01235-f020:**
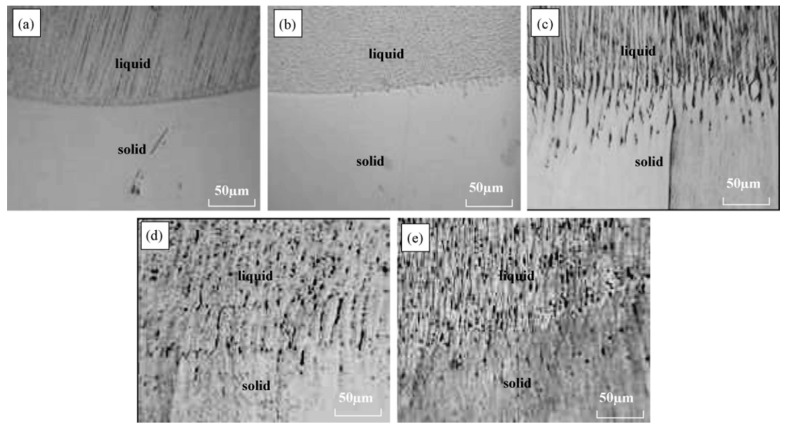
Microstructure at the pulling rate of 12 μm/c with different magnetic intensity: (**a**) 0 T, (**b**) 0.50 T, (**c**) 0.84 T, (**d**) 1.00 T, and (**e**) 1.50 T [21].

**Figure 21 materials-15-01235-f021:**
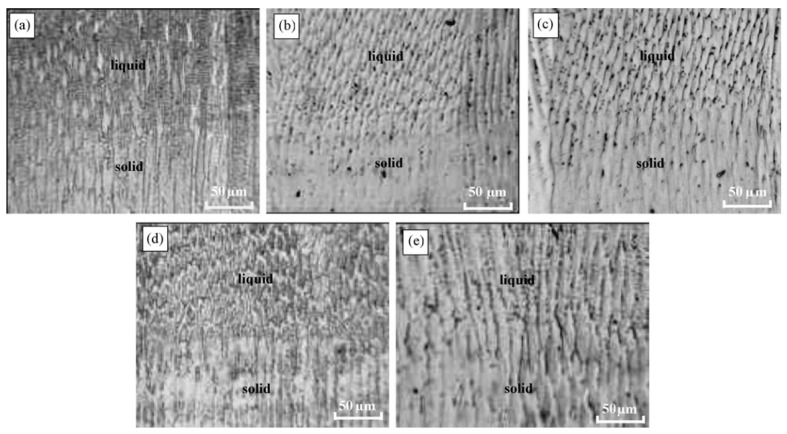
Longitudinal microstructure at the pulling rate of 96 μm/c with different magnetic intensity: (**a**) 0 T, (**b**) 0.50 T, (**c**) 0.84 T, (**d**) 1.00 T, and (**e**) 1.50 T [21].

**Figure 22 materials-15-01235-f022:**
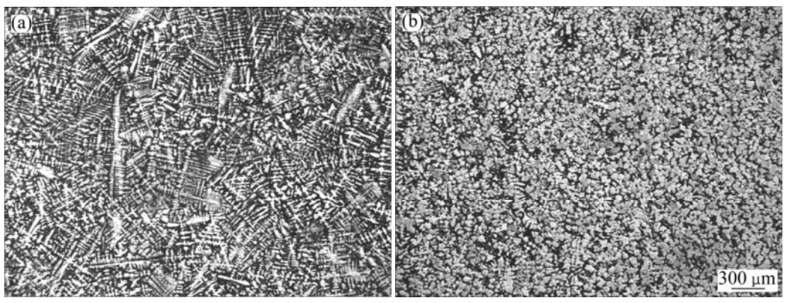
As-cast microstructures of superalloy IN718 without PMF (**a**) and with PMF (**b**) [22].

**Figure 23 materials-15-01235-f023:**
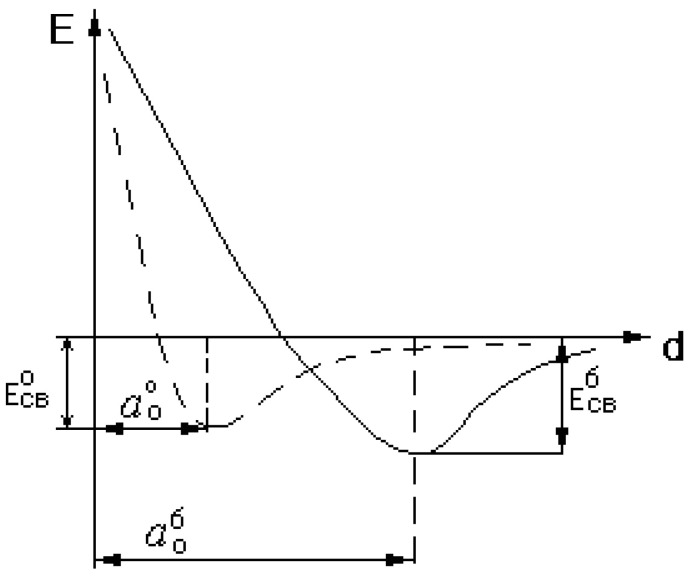
Changes in the binding energy during the approach of atoms in liquid metals under the influence of EMP: a^0^_0_ and a^b^_0_—interatomic distances in initial and treated liquid metals, respectively; E^δ^_CB_ and E^0^_CB_ are the binding energies in initial and treated liquid metals [47].

**Figure 24 materials-15-01235-f024:**
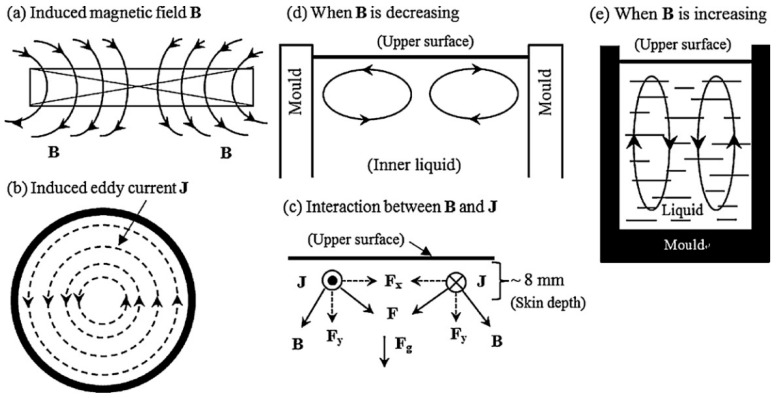
Illustration of fundamental principles of the PMF process: (**a**) side view of magnetic field induced by the induction coil; (**b**) top view of induced eddy current in the melt; (**c**) interaction between the magnetic field and eddy current; (**d**) a minor form of forced convection within the surface layer occurs when B is decreasing; and (**e**) a major form of forced convection is aroused when B is increasing [23]. Copyright 2012, Elsevier.

**Figure 25 materials-15-01235-f025:**
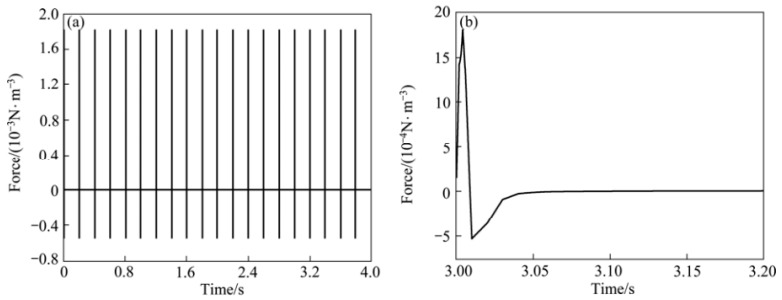
Curves of magnetic force of node at external center of melt: (**a**) 20 whole periods; and (**b**) change of magnetic force in upward period and falling period of pulse [22].

**Figure 26 materials-15-01235-f026:**
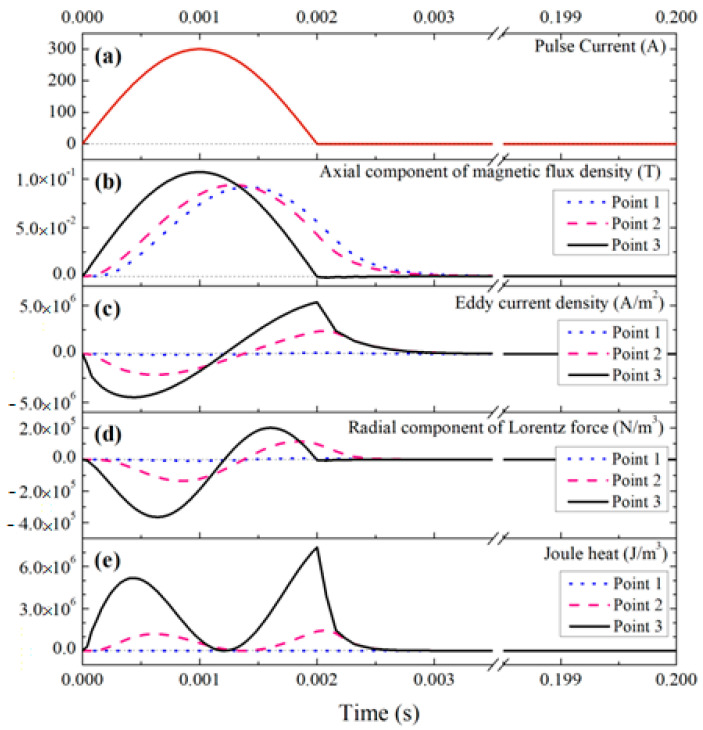
Results of modeling (**a**) the current of the exciting pulse in the coil; (**b**) the axial component of the magnetic flux density; (**c**) eddy current density; (**d**) the radial component of the Lorentz force; and (**e**) Joule heat at points 1–3 (1 cent of the melt, 2 half the radius, and 3 at the wall of the mold) [100].

**Table 1 materials-15-01235-t001:** Characteristics of generators.

Type	Amplitude, kV	Duration, ns	Frequency, kHz	Front, ns	Resistance, Ω
Generators IEF	30–450	10–40	0,1	-	200–300
Generators IYaF	5–10	1	1	0.1	50
Generators FID	1–100	0.1–10	0.1–100	0.01–1	0.2–1000

**Table 2 materials-15-01235-t002:** Characteristics of the Cu-37.4 wt.% Pb after pulse treatment [37].

Properties	I = 0 A	I = 800 A	I = 1000 A	I = 1200 A	I = 1400 A
Hardness, kgf·mm^2^	34.3	40.5	42.7	43.4	49.3
Friction coefficient	0.24	0.23	0.21	0.20	0.18
Wear rate	1.3	1.0	0.9	0.7	0.4

**Table 3 materials-15-01235-t003:** Results of mechanical testing of steel 35L [45].

Sample	YS, (MPa)	TS, (MPa)	δ, (%)	ψ, (%)	KCU^+20^, (J/cm^2^)	HRC
«Unirradiated»	386	520	6	12	49	22
«Irradiated»	467	780	20	39	50	25

**Table 4 materials-15-01235-t004:** The effect of the duration of the EMP treatment on the corrosion of cast iron [48].

	0 min	5 min	10 min	15 min	20 min
Δm/S, g/sm^2^	0.07	0.06	0.04	0.035	0.07
KcoH_2_. sm^2^·h	3.1	2	1.5	1	3.5

**Table 5 materials-15-01235-t005:** Influence of PMP parameters on structure and properties of hardeners Al-20%Si [51].

Type of Processing	Number of Pulses	Average Size of Sip Cystals, μm	Density, g/sm^3^	Electric Conductivity, MS/m
Without processing	-	530	2.658	13.2
PMP	1	282	2.670	14.7
	2	106	2.679	15.1
	3	88	2.681	15.2

**Table 6 materials-15-01235-t006:** Efficiency of transformation of fields in metals [17].

Metal	β	W2	W1
Al (T = 300 K)	0.033	1	0.033
Na	0.75	4.6	3.5
K	1.81	2.4	4.3
Al	0.88	1.1	0.97
Cu	0.88	0.3	0.26
In	3.1	0.1	0.31
Cs	11.8	0.07	0.83
Pb	15.1	0.004	0.06
Bi	23.5	0.002	0.05
Hg	21	0.0019	0.04

**Table 7 materials-15-01235-t007:** Parameters of impulse processing of melts.

Type of Processing	Duration, (s)	Frequency, (Hz)	Amplitude of Current, (A)	Amplitude of Voltage, (kV)	Pulse Power, (W)	Average Power, (W)
ECP	0.0024–0.011	1–200	300–2200	2.5–3	-	-
EMP	(0.1–40) × 10^−9^	100–1000	-	1–15	2 × 10^9^	2
PMF	0.003–0.23	2–7000	650–50000	1.5–2.5	-	-

**Table 8 materials-15-01235-t008:** Pulse processing parameters.

Source	Amplitude of Current, (A)	Leading Edge of a Pulse, (ms)	Δj/Δt, (A/s)
[1]	1200–2200	0.6–2.75	3.6 × 10^6^
[3]	600	5	1.2 × 10^5^
[7]	300	1 × 10^−6^	3 × 10^12^
[11]	120, 200	1 × 10^−6^	1.2 × 10^12^; 2 × 10^12^
[20]	3000	1 × 10^−6^	1.5 × 10^7^
[21]	650	6.5	1 × 10^5^
[22]	50000	0.8	6.2 × 10^7^

**Table 9 materials-15-01235-t009:** Effects of impulse exposure.

#	Effect	Source		
		ECP	EMP	PMF
1	Changing the crystal structure of the ingot (increasing the proportion of equiaxed grains), grinding macrograins	[3,4,35,37,52,53,57,58,59]	[13,15,18]	[49,22,23,50,51]
2	Reducing the degree of supercooling of the melt	[2]	-	[49]
3	Increase in liquidus temperature	[37]	[38]	[100]
4	Change in the quantitative ratio of equilibrium phases	[36]	[13,15,43,44]	-
5	Refinement of micrograins	[36]	[13,15,47]	[50,51]
6	Increase in yield strength, increase in elongation	[35]	[13,15,18,45]	[51,57]
7	Reduced coefficient of friction and wear rate	-	[37]	-
8	Increasing macro/microhardness	[36,37]	[13,15,18,45,46]	-
9	Decrease in electrical resistivity	-	[18]	[51]
10	Increased corrosion resistance	[36]	[48]	-
11	Increasing the solubility of the second component in the main phase	[37]	[15,18]	
12	Reducing the number of inclusions and dispersed particles of the second phase	[38,39,40,41,42]	[43,44]	
13	Increase in density, decrease in porosity	-	[18,43,46,47]	[23,51]

**Table 10 materials-15-01235-t010:** Impulse action mechanisms.

#	Mechanism of Action	Source		
		ECP	EMP	PMF
1	The emergence of force flows in the melt, which cause fragmentation of the formed dendrites and increase the number of crystallization centers	[2,28,58]	[12,13,14,15,16,17,18,20]	[22,49,51]
2	Changing the nucleation mechanism due to a decrease in the free energy of the system	[52,53]	-	[21]
3	Cluster theory, according to which an external field makes clusters of one type more stable than another	[37]	[11,47]	-
4	Additional heating of the melt and thermal vibrations	-	[12,13,14,15,16,17,18]	[23,51]
5	Appearance in the melt under the influence of external action of forces of a mechanical nature (vibrations, vibration, ultrasonic vibrations)	[28]	[12,13,14,15,16,17,18,44]	[23,49,51]

## Data Availability

No new data were created or analyzed in this study. Data sharing is not applicable to this article.

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
