# Peer review of "Theory and Practice of Using Pulsed Electromagnetic Processing of Metal Melts"

_materials, 2022, doi:10.3390/ma15031235_

Round 1

Reviewer 1 Report

Please see the attached report.

Author Response

Dear Reviewer!

First of all, we want to thank you for your attention and interest in our work. We also want to express our gratitude to you for your comments on the article. Working on them allowed us to significantly improve our article.

All changes in the text are highlighted and marked.

 Below are short responses to your comments.

(1) The connection or difference between studies performed by researches should be given rather than simply shown. For example, 3.1 presents the results of the pulsed electric current applied in metal alloys. Many researches have been given by only showing the main results of the work. However, it is difficult to receive the connection or progress of the research in comparison with the previous studies. It is no doubt that the grain refinement in solidifying metal alloys driven by pulsed electric currents have been intensively investigated. But, this topic has been mixed with other topics, such as directional solidification (crystal growth) and macrosegregation (phase distribution) influenced by electric currents. These topics should be separately reviewed rather than mixed together in the same paragraph. In a word, this is review, not an experimental report.

Answer. You recommend not just stating the results of the authors' research, but showing them in the dynamics of development. This is certainly a very correct recommendation, but, unfortunately, it is difficult to apply in the case of impulse processing. Because in many respects all works are identical. Starting from the type of alloys used and ending with the equipment used. Therefore, it was important for us to show the experimental and theoretical material accumulated to date, so that it would be easier for the experimental reader to plan future experiments without repeating those already done. For a detailed analysis and systematization of the presented data, we have provided for Section 4, in which we have brought together and systematized all the information presented. We very much hope that our work will be interesting to readers and convenient for practical application.

(2) Some new or important publications should be added in this review. For example, a pulsed magnetic field was firstly employed by Zi et al. [1] to provoke the grain refinement in solidified metal alloys. This work even can be regarded as a milestone for the application of pulsed electromagnetic fields in solidified metal alloys, because this contactless way to influence the solidification process of metal alloys can be well accepted in industry. Moreover, Jie et al. [2] recently published a paper with respect to the grain refinement in commercial pure aluminum under the impact of a pulsed magnetic field. A new grain refinement mechanism has been proposed. Hence, these two publications should be cited. Of course, other new and important publications should be carefully screened and cited in this review. [1] B.T. Zi, Q.X. Ba, J.Z. Cui, Y.G. Bai, X.J. Na, Effect of strong pulsed electromagnetic field on metal's solidified structure. Acta Phys. Sin. 49, 1010 (2000). [2] Jie, J.C., Yue, S.P., Liu, J., StJohn, D.H., Zhang, Y.B., Guo, E.Y., Wang, T.M., Li, T.J. Revealing the mechanisms for the nucleation and formation of equiaxed grains in commercial purity aluminum by fluid-solid coupling induced by a pulsed magnetic field (2021) Acta Materialia, 208, 116747. (3) Many typos and citations should be carefully checked and corrected. These typos should be revised.

Answer. Many thanks for the noted errors in references to literary sources. We have corrected and rechecked them.

We also added a number of publications, as you recommended. But, unfortunately, not all publications are in the public domain. For example, the text "B.T. Zi, Q.X. Ba, J.Z. Cui, Y.G. Bai, X.J. Na, Effect of strong pulsed electromagnetic field on metal's solidified structure. Acta Phys. Sin. 49, 1010 (2000)" publication could not be found. Authors of this article on text request articles in Researchgate also did not respond. Similar problems arose when searching for the main material for the article. For example, the work "Q.-S. Li, L.-Q. Liu, Q.-J. Zhai. Analysis of graphite morphology of gray cast iron in pulse magnetic field. Journal of Iron and Steel Research International. 2005. (12) 2, pp. 45 - 48 "was also of interest to us, but we could not find its text. We found several papers that refer to the article you proposed (B.T. Zi, Q.X. Ba, J.Z. Cui, Y.G. Bai, X.J. Na, Effect of strong pulsed electromagnetic field on metal's solidified structure Acta Phys Sin 49, 1010 (2000) and realized that the results of her, if not repeating, are in good agreement with that in our article. We are very sorry that we could not fully implement your recommendations.

Hope for your understanding.

Best regards,

Authors Team

Reviewer 2 Report

The article is a review of metal melting through the use of pulsed electric, magnetic, and electromagnetic fields. It may be considered a very interesting reading for researchers intending to start in these domains, but it is also a good reference for those that actually are already used to these themes. A vast number of techniques are described in the text and may certainly be handy for experimentalists. The number of references as well as the diversity of authors seem to be correctly distributed.

It appears also clear that the paper has been written at many hands, some of them more careful than others, especially regarding style, format, and grammar. I would recommend the authors to be very vigilant to the correct use of the first person while writing. General knowledge must be described as a passive voice, i.e., impersonally, while the first person, when and if used, is strictly reserved to the very specific point being addressed.

Different parts of the text are particularly colloquial, which must be avoided in scientific reports. Equally, the format editing deserves special attention in respect of all the readers around the world. I strongly recommend that the authors take time to improve their editing. To communicate scientific results is important, but to do it properly is as much important as well.

Author Response

Dear Reviewer!

First of all, we want to thank you for your attention and interest in our work. We also want to express our gratitude to you for your comments on the article. Working on them allowed us to significantly improve our article.

All changes in the text are highlighted and marked.

Best regards,

Authors Team

Reviewer 3 Report

Dear Authors,

The paper is generally well written. However, several comments should be addressed and solved in revised version of the manuscript. Please follow the points listed below:

  1. An abstract is too short. It should be extended.
  2. Section. 1. Introduction - should be expanded is poorly described - or maybe little joined with section 2
  3. Please improve the quality of several Figures: 7,13, - seem t be fuzzy, 26 - please add y-axis caption
  4. Sec. 2.1. Line 42 - please add equation
  5. Table 1 - I think that it better is to express standardized captions of physical quantities like - Resistance, Ohm (instead of Load) etc. 
  6. Table 7 and others - please use dots as decimal separators,
  7. Eq14 - please check this equation and use proper symbols
  8. Conclusions - this section should be more essential. Maybe it would be better to divide it for discussion and then conclusions.

Author Response

Dear Reviewer!

First of all, we want to thank you for your attention and interest in our work. We also want to express our gratitude to you for your comments on the article. Working on them allowed us to significantly improve our article.

All changes in the text are highlighted and marked.

 Below are short responses to your comments.

  1. An abstract is too short. It should be extended.

Answer. Thank you, we have corrected the text of the manuscript.

  1. Section. 1. Introduction - should be expanded is poorly described - or maybe little joined with section 2

Answer. We have updated the text.

  1. Please improve the quality of several Figures: 7,13, - seem t be fuzzy, 26 - please add y-axis caption

Answer. We have corrected figures 7 and 13. 26 is taken from the source [100], the values ​​of the "y" axes are signed in the field of graphs.

  1. Sec. 2.1. Line 42 - please add equation

Answer. We have updated the text.

  1. Table 1 - I think that it better is to express standardized captions of physical quantities like - Resistance, Ohm (instead of Load) etc.

Answer. Thank you, we have corrected the text.

  1. Table 7 and others - please use dots as decimal separators,

Answer. Thank you, we have corrected the text.

  1. Eq14 - please check this equation and use proper symbols

Answer. The equation has been verified. Corresponds to the original.

  1. Conclusions - this section should be more essential. Maybe it would be better to divide it for discussion and then conclusions.

Answer. Thank you for your comment. Corrections have been made to the text.

Best regards,

Authors Team

Round 2

Reviewer 1 Report

I think the present version can be accepted for publication.

Reviewer 3 Report

Dear Authors,

The paper can be considered for publication. I checked all addressed issues and comments. All comments and suggestions were replied and solved. The paper is now improved.